

# Impacts of Current and Projected Oil Palm Plantation Expansion on Air Quality Over Southeast Asia

Sam J. Silva[1], Colette L. Heald[1], Jeffrey A. Geddes[2], Kemen G. Austin[3], Prasad S. Kasibhatla[3], Miriam E. Marlier[4]

[1]Department of Civil and Environmental Engineering, Massachusetts Institute of Technology, Cambridge, MA, USA
[2]Department of Physics and Atmospheric Science, Dalhousie University, Halifax, Nova Scotia, Canada
[3]Nicholas School of the Environment, Duke University, Durham, NC, USA
[4]Department of Ecology, Evolution and Environmental Biology, Columbia University, New York, NY, USA

*Correspondence to*: Sam Silva (samsilva@mit.edu)

**Abstract.** Over recent decades oil palm plantations have rapidly expanded across Southeast Asia (SEA). According to the United Nations, oil palm production in SEA increased by a factor of 3 from 1995 to 2010. We investigate the impacts of current (2010) and future (2020) oil palm expansion in SEA on surface-atmosphere exchange and the resulting air quality in the region. For this purpose, we use satellite data, high-resolution land maps, and the chemical transport model GEOS-Chem. Relative to a no oil palm plantation scenario (~1990), overall simulated isoprene emissions in the region increase by 13% due to oil palm plantations in 2010 and a further 11% by 2020. In addition, the expansion of palm plantations leads to local increases in ozone deposition velocities of up to 20%. The net result of these changes is that oil palm expansion in SEA increases surface $O_3$ by up to 3.5 ppbv over dense urban regions, and could rise more than 4.5 ppbv above baseline levels by 2020. Biogenic secondary organic aerosol loadings also increase by up to 1 µg m$^{-3}$ due to oil palm expansion, and could increase a further 2.5 µg m$^{-3}$ by 2020. Our analysis indicates that while the impact of recent oil palm expansion on air quality in the region has been significant, the retrieval error and sensitivity of the current constellation of satellite measurements limit our ability to observe these impacts from space. Oil palm expansion is likely to continue to degrade air quality in the region in the coming decade and hinder efforts to achieve air quality regulations in major urban areas such as Kuala Lumpur and Singapore.

## 1. Introduction

Palm Oil is currently the most popular source of food oil in the world, and is rapidly gaining importance as a source of biofuel (Corley, 2009). Over 80% of global palm oil production takes place in Southeast Asia (SEA), with more than 10 million hectares of land farmed in 2005 (Fitzherbert et al., 2008). According to the Food and Agriculture Organization of the United Nations (faostat3.fao.org), palm oil production in SEA has grown by a factor of 5 over the past 20 years. By 2020, the total area occupied by oil palm plantations in SEA is expected to increase even further (Austin et al. 2015, Marlier et al. 2015b). This represents a significant and rapid change in land use over a relatively small region. The palm expansion in SEA



is just one example of land use change that may drive changes in atmospheric composition; however, to date there has been limited exploration of these impacts (Heald and Spracklen, 2015).

The impact of oil palm expansion on global carbon stocks and biodiversity has been studied extensively. Austin et al. (2015) suggest that up to 47 MtC yr$^{-1}$ will be emitted due to oil palm land use change in Kalimantan alone from 2010-2020. Carlson et al. (2013) estimate that the plantations could generate over 25% of Indonesia's projected carbon emissions by 2020. Furthermore, replacing native forests with oil palm has and will continue to threaten biodiversity across the region. Fitzherbert et al. (2008) found that only 15% of all species present in the natural forest were also found in oil palm plantations. This rapid land use change across SEA has also altered surface-atmosphere exchange of trace gases over the region, with potential implications for regional air quality. Airborne observations over northeastern Borneo during the 2008 OP3 campaign indicated that oil palm plantations emit 7 times more isoprene than the nearby rainforest (Hewitt et al., 2010). Isoprene is the most abundant volatile organic compound (VOC) emitted by plants, is a precursor of secondary organic aerosol (SOA), and can play an important role in surface ozone formation, depending on the local chemical environment. Despite this dramatic increase in isoprene emissions, no appreciable difference in surface $O_3$ over palm versus native forest was observed during OP3 (Hewitt et al., 2009). Growth of oil palm plantations also impacts the deposition of gases and particles from the atmosphere due to changes in the total amount of plant surface area available for deposition. The OP3 observations show that deposition velocities of ozone over oil palm are half the magnitude of deposition velocities over the natural forest (Fowler et al., 2011). Fowler et al. (2011) suggest that this is due to smaller non-stomatal exchange of ozone from the oil palm canopy. Expansion of oil palm may also impact soil $NO_x$ emissions due to fertilizer application, changing land types, changes in the amount of NO that escapes through the canopy, and changes in the amount of sunlight that reaches the ground (Hewitt et al., 2009). The links between these rapid land use changes and air quality become even more important in light of the tremendous population growth and significant air quality problems already present throughout SEA. The region contains more than 570 million people, an increase of more than 10% since 2000 (United Nations Population Division, 2011). Forouzanfar et al. (2015) found that indoor and outdoor atmospheric pollution are both within the top 10 leading risk factors for premature mortality throughout SEA.

The impact of oil palm plantations on air quality has been investigated in previous modeling studies. Warwick et al. (2013) simulate the influence of modern day oil palm distributions over Borneo, constraining both isoprene and $NO_2$ fluxes to values measured during the OP3 campaign. They found that the addition of oil palm in Borneo contributed to a local increase in $O_3$ concentrations on the order of 30% (~8 ppbv). Moreover, they suggest that converting the entire island of Borneo to palm plantations could lead to a 70% (~18 ppbv) increase in surface $O_3$ concentrations. Ashworth et al. (2012) use the HadGEM2 model to assess the impact of biofuel feedstocks on air quality. Part of their study included oil palm plantations in SEA, where they found that the increase in isoprene emissions by 2020 could lead to local increases in surface $O_3$ and SOA of up to 3 ppbv and 0.4 μg m$^{-3}$, respectively.



Agricultural burning across SEA (partially related to palm oil production) is an additional source of air pollution in the region. Marlier et al. (2015b) assessed different land use scenarios across Sumatra to determine their overall influence on current and future fire emissions. They found that the scenario with the highest amount of oil palm had the largest associated fire emissions (100 Tg DM yr$^{-1}$); these fires contribute up to 60% of the total smoke concentrations across equatorial SEA.

In this study, we use the GEOS-Chem model to simulate the potential impacts of oil palm on air quality across SEA for current and future oil palm expansion scenarios. We go beyond previous studies by explicitly simulating the concurrent perturbations to biogenic emissions, soil NO$_x$ emissions, and dry deposition, and exploring the net impacts on air quality in the region. We compare these results with available satellite observations to investigate whether the current constellation of satellite instruments can detect the changes in air quality driven by rapid land use change.

## 2. Model Description

### 2.1 The GEOS-Chem model

We use the global chemical transport model GEOS-Chem v9-02 (www.geos-chem.org) to investigate the changes in air quality associated with oil palm plantations in SEA. The model is driven by assimilated meteorology from the Goddard Earth Observing System (GEOS). We use GEOS-5 meteorology for the year 2006 in all of our simulations, due in part to the availability of updated anthropogenic emission inventories for this year. For this analysis, we perform a series of nested simulations of GEOS-Chem over the Asian domain (70˚–150˚E, 10˚S–55˚N) at 0.5˚x0.667˚ horizontal resolution with 47 vertical layers. Boundary conditions are produced using the same version of the global model at 2˚x2.5˚ horizontal resolution. The model was initialized with a 1-year simulation at 2˚x2.5˚ horizontal resolution, and then an additional 6-month simulation at 0.5˚x0.667˚.

The GEOS-Chem oxidant-aerosol simulation includes H$_2$SO$_4$-HNO$_3$-NH$_3$ aerosol thermodynamics (Park et al., 2006; Pye et al., 2009) coupled to a detailed HO$_x$-NO$_x$-VOC-O$_3$-BrO$_x$ chemical mechanism (Bey et al., 2001; Mao et al., 2013). Secondary organic aerosol (SOA) is produced from the oxidation of biogenic hydrocarbons (isoprene, monoterpenes, and sesquiterpenes), aromatics, and IVOCs and represented with a volatility basis set approach (Pye et al., 2010; Pye and Seinfeld, 2010).

The global simulations are driven by anthropogenic emissions from the Emissions Database for Global Atmospheric Research version 3 (EDGARv3), including emissions from ship exhaust (Olivier et al., 2001). Over the Asian region, a more recent anthropogenic emission inventory from the year 2006 is used (Streets et al., 2003, 2006). Geddes et al. (2015b) show that the long-term (1996-2012) trend in satellite-derived estimates of ground-level NO$_2$ concentrations over SEA is relatively



small, indicating that regional changes in anthropogenic emissions are not large. Global emissions from aviation are based on the AEIC inventory (Stettler et al. 2011, Simone et al. 2012). Biomass burning emissions for 2006 follow the GFED3 inventory (van der Werf et al., 2010). We note that the GFED3 emissions indicate that 2006 was a higher than average fire year in the region.

The GEOS-Chem land use module developed by Geddes et al. (2015a) is used to drive surface-atmosphere exchange processes in the model. Land use is described using 16 plant functional types (PFTs), with an associated monthly leaf area index (LAI) per PFT. The baseline global PFTs and associated LAI are from the year 2000 inputs to the Community Land Model v.4 (http://www.cgd.ucar.edu/tss/clm), which are based on satellite observations (Lawrence et al., 2011). The

dominant vegetation types in SEA are plotted in Figure 1. Biogenic emissions of VOCs (BVOCs) are calculated online for each PFT following MEGANv2.1 (Guenther et al., 2012). These BVOC emissions respond to temperature, available sunlight, leaf area index, leaf age, and soil moisture. These responses are quantified as activity factors which are applied to the basal emission factor to calculate emissions that vary with meteorology and phenology. Dry deposition is calculated following Wesely (1989). The depositional velocity is a function of the aerodynamic, boundary layer, and canopy

resistances, added in series. The aerodynamic resistance depends on atmospheric stability and surface roughness height, the boundary layer resistance is a function of the chemical species and meteorology, and the canopy resistance varies with the chemical properties of the deposited species and the land type. To calculate the canopy resistance, land types are mapped from the 16 PFTs to the 11 depositional surfaces described in Wesely (1989). For the calculation of the aerodynamic and canopy resistances, we take into account the influence of LAI on the land properties. We account for all depositional surface

types within a grid cell by preserving the fractional land cover of each grid box. Soil $NO_x$ emissions are a function of soil moisture, temperature, available nitrogen, and land use type and are calculated following Hudman et al. (2012). The land types for these emissions are mapped from the PFTs to 24 biomes described in Steinkamp and Lawrence (2011). Additionally, a canopy reduction factor, to account for the loss of NOx within the canopy, is calculated as a function of LAI and meteorological parameters.

GEOS-Chem has previously been used to study air quality in SEA. Trivitayanurak et al. (2012) use the nested model to better understand the distribution and sources of atmospheric trace constituents over Asia. They find that the model captures the vertical and spatial variability of trace constituents such as CO, isoprene, and sulfate to within 30% of observations from several aircraft campaigns. They also note that the model under predicts AOD as measured by MODIS, which they attribute

to an underestimate of local biogenic SOA. Fu et al. (2007) use the model in combination with satellite measurements of formaldehyde to constrain non-methane VOC emissions broadly across Asia from 1996-2001. They show that observed and modeled HCHO are highly correlated in the region. Jiang et al. (2015) compare the model and satellite estimates of CO, $O_3$, and $NO_2$ in Asia, and conclude from these comparisons that the GEOS-Chem simulation of tropospheric $O_3$ is reliable within the Asian domain.



## 2.2 Description of SEA land use

To account for oil palm plantations, we add a new plant functional type to the land module to calculate palm specific biogenic emissions, soil $NO_x$ emissions, and dry deposition. The basal isoprene emission factor of oil palm is set to the basal rate of 7.8 mg $m^{-2}$ $h^{-1}$ measured during OP3 (Misztal et al., 2011). The basal isoprene emission factor of the native forests is

reduced to 1.6 mg $m^{-2}$ $h^{-1}$ to match the OP3 measurements (Langford et al., 2010), which is a factor of 4 lower than the emission factors for broadleaf evergreen tropical trees within MEGANv2.1, and consistent with the rainforests of Southeast Asia emitting less isoprene than South American and African rainforests. This difference is likely due to a previous dearth of measurements across the rainforests of SEA to constrain MEGANv2.1 (Guenther et al., 2006; Guenther et al., 2012). Observations suggest that that oil palm is not a significant emitter of monoterpenes and sesquiterpenes, but the mechanisms

behind this are not clear (Misztal et al., 2011). In light of this, and the large uncertainties on the estimates of basal emissions for monoterpenes and sesquiterpenes for natural forests in the region, we modify only the emission factor of isoprene for oil palm relative to background forests.

During the OP3 campaign Fowler et al. (2011) observed that relatively few of the oil palm plantations were fertilized. When

extrapolated across the whole expanse of oil palm plantations, it was found that the average soil $NO_x$ emission of plantations is similar to that of the background forest. In addition, we have no information on the relative impacts of palm plantations and natural tropical forests on nitrogen cycling in the soil. Therefore, the biome type-specific soil $NO_x$ emissions parameters for oil palm are identical to the tropical forest conditions in our simulations.

The leaf area index (LAI) of an oil palm plantation changes as the trees age. Oil palms themselves live for more than a decade, and their leaves live for 600-700 days; this contributes to LAI values that can vary from 2 to 8 in mature plants (Van Kraalingen et al., 1989). LAI is set to 4.5 for these simulations; this value is selected as an approximation of the average over the lifespan of the plant (Van Kraalingen et al., 1989), and therefore a likely average for the plantation as a whole. We explore the sensitivity of our results to this specified LAI value in Section 3. For the Wesely depositional scheme land types,

oil palm is assumed to be most similar to the native forest, as opposed to the "crop" land type. This assumption is made simply because oil palm plantations are physically much closer to a tall tree forest than a wheat field or cornfield. We do not attempt to model the detailed growth, harvesting, and senescence history of the plantation (Fan et al., 2015).

Several land use maps are used to describe the modern and future distribution of oil palm over SEA (Figure 2). For the

modern day scenario, we use a land use map developed by Miettinen et al. (2012), which describes land use across insular SEA (-10˚ N to 10˚N, 95˚E to 140˚ E) in 2010 on a 250 meter grid, and includes a land classification for palm plantations. The future expansion scenario maps are adapted from Marlier et al. (2015b) and Austin et al. (2015). Marlier et al. (2015b) developed a variety of scenarios at 1km resolution to understand changes in fire emissions associated with land use change



in Sumatra. We use the "High oil palm" scenario to represent a realistic upper limit on the 2020 distribution of Sumatran oil palm. This map was reported originally as the probability that a given grid box will contain oil palm in 2020. The probability for each grid box was treated as the percent area covered by palm, and converted to fractional PFT coverage at the 0.23˚x0.31˚ resolution that is input into the GEOS-Chem land module. This leads to an increase of 112% in total palm coverage in Sumatra from 2010 to 2020. Austin et al. (2015) mapped the future oil palm distribution in Kalimantan (Indonesian Borneo) using a logistic regression model at 250 meter resolution to understand the impact that oil palm could have on emissions of $CO_2$. We constrained that logistic regression model to a total of 3.6Mha of oil palm expansion to create a map for the 2020 distribution of oil palm in Kalimantan, following the totals given by Austin et al. (2015). This leads to an increase of 108% in total palm coverage over Kalimantan from 2010 to 2020. Note that these two datasets cover only Sumatra and Kalimantan, which together represent 54% of palm production in the region in 2010, but much of the available land for future expansion. From 2010 to 2020 oil palm production in these two regions increase by 10.8 Mha. Our projections do not consider the potential oil palm expansion in the rest of the SEA region. For both scenarios, the oil palm plantations are added as a land type and fractional coverage of pre-existing land classes from the base land map are reduced accordingly.

## 3. Impacts of palm expansion on air quality in Southeast Asia

To understand the overall air quality impact of oil palm plantations in SEA, we explore three major plantation scenarios. The first uses a land map with no palm to establish a baseline. This simulation is referred to as "No Palm" and is representative of conditions that pre-date the major palm expansion in SEA (~1990). The next is a simulation we call "Modern Palm", using the 2010 distribution from Miettinen et al. (2012). The third simulation is "Future Palm," and uses the merged land maps from Marlier et al. (2015b) and Austin et al. (2015) for Sumatra and Kalimantan. In addition, we perform sensitivity scenarios for the Modern Palm distribution to disaggregate land use change-driven impacts on BVOC emissions from dry deposition and soil NOx emissions; this is referred to as "BVOC-only". Results are shown here for annual means; seasonal differences are minor.

### 3.1 Changes in Surface-Atmosphere Exchange

The direct influence of oil palm plantation expansion is on surface-atmosphere exchange, most significantly BVOC emissions and dry deposition.

The changes in BVOC emissions are as anticipated: where palm is added, surface fluxes of BVOCs increase. By far the largest increase is that of isoprene. There are increases in other BVOCs due to replacement of unforested regions (wetlands, pastureland, etc.) with oil palm, but they are at least a full order of magnitude smaller than the concomitant changes in isoprene. Figure 3 shows the increase in isoprene emissions associated with the addition of oil palm plantations in both the





Modern and Future scenarios, compared to the baseline No Palm scenario. The largest increases in isoprene emissions in the Modern Palm scenario occur in northern Borneo, Sumatra, and the southern Malay Peninsula. This addition of palm increases isoprene emissions by a factor of three over northeastern Borneo, where the OP3 campaign took place (see section 4). This growth corresponds to an absolute increase on the order of $1.2 \times 10^{12}$ atoms C $cm^{-2}$ $s^{-1}$. The largest relative changes in

5    isoprene emissions occurred over the northern half of Sumatra and the southern Malay Peninsula, with up to a 4.5 fold increase over the simulation without oil palm. Similar to northeastern Borneo, the absolute change in emissions is on the order of $10^{12}$ atoms C $cm^{-2}$ $s^{-1}$. Sumatra and the Malay Peninsula are of particular importance due to their proximity to the large urban centers of Kuala Lumpur and Singapore. Oil palm plantations in 2010 result in an additional 1.26 TgC $yr^{-1}$ of isoprene emission from SEA, a 13% increase from the No Palm scenario.

The Future Palm emission scenario changes are limited to Sumatra and Kalimantan (Indonesian Borneo) as prescribed in the land use change scenarios considered (Section 2.2). Isoprene emissions in Kalimantan increase by a factor of 2 ($\sim 0.9 \times 10^{12}$ atoms C $cm^{-2}$ $s^{-1}$) from the No Palm simulation. These are mostly regions that are as yet undeveloped and are good candidates for future palm agriculture. There are many regions in Sumatra where the changes in isoprene emissions are

15    greater than a factor of 3-4, with an absolute difference in excess of $1.2 \times 10^{12}$ atoms C $cm^{-2}$ $s^{-1}$. Similar to the changes in the Modern Palm scenario, these isoprene emissions increases are very near large urban regions. In this scenario, the total change in isoprene emissions from 2010 to 2020 in SEA is 1.1 TgC $yr^{-1}$ across SEA, a 10% increase in isoprene emissions. This emission increase is nearly twice as high as previous work (Ashworth et al., 2012), due to differences in the assumed future distribution of oil palm.

The impacts of oil palm expansion on dry deposition velocities are more complex than the BVOC emissions changes. As stated in Section 2.2, LAI influences the resistance terms in the calculation of depositional velocities, and oil palm plantations tend to have a higher LAI (prescribed here to a fixed value of 4.5) than both the natural forest (ranges from 0 to 6.77, with a median value of 4 for SEA), and grasslands or previously cleared agricultural lands. The LAI for the various

25    scenarios is shown in Figure 4. The addition of oil palm plantations increases the LAI for much of SEA. For a highly reactive species such as $O_3$, an increase in LAI directly leads to an increase in the depositional velocity over that surface, as seen in Figure 5. It should be noted that there are minor perturbations to the deposition of other gas phase species as well. However, $O_3$ deposition is most sensitive to this land use change due to its high reactivity and strong LAI dependence. The impact of oil palm expansion on particle deposition is negligible.

Modern palm distributions increase $O_3$ dry deposition velocities most significantly in the Malay Peninsula and North Sumatra. In both regions, the largest changes are increases of 0.05 cm $s^{-1}$, or nearly 15%. This change is the opposite sign and smaller in magnitude than the measured difference in $O_3$ deposition velocity across the forest to palm transition reported by Fowler et al. (2011); however our values are not directly comparable to those measurements given the heterogeneity of





land types within each grid cell, the resolution of the model, and the fact that the simulated depositional velocity changes are an aggregate of many land types transitioning to oil palm plantations (not purely forest to palm). Figure 4 shows that the Malay Peninsula and North Sumatra exhibit the largest changes in deposition partially due to the dense palm plantations in the region, and also due to the large changes in LAI due to those plantations replacing cleared land, low LAI forests, and

other varied land types. Across SEA, the overall impact of palm plantations on ozone deposition is small, with a net 0.5% increase in $O_3$ depositional velocity.

The future distribution of oil palm in 2020 produces the largest changes in ozone dry deposition velocities over Kalimantan, and again Sumatra (Figure 5). In both these regions, increases in deposition velocities are as large as 0.06 cm s$^{-1}$, or 20%

relative to the No Palm scenario. As with the Modern Palm scenario, the most significant changes occur over areas where oil palm plantations replace cleared lands. The average change in ozone dry deposition fluxes across the region from no Palm to 2020 is an increase of 1.0%.

In this work, the basal emission of soil $NO_x$ in oil palm plantations is identical to that in natural forests. However the loss of

$NO_x$ to the canopy is impacted by differences in LAI. We find that there are negligible changes in the net soil $NO_x$ emission over regions where the land transitioned from forest to palm. This is largely consistent with the OP3 field campaign observations (Fowler et al., 2011), which indicated that few of the palm plantations are fertilized. The soil $NO_x$ changes are more significant where other land type changes occurred; for instance, moving from pastureland to oil palm. However there remains substantial uncertainty surrounding fertilization practices in SEA palm plantations. Therefore, this simulated

perturbation in soil $NO_x$ emissions is likely a lower limit. Additional small changes in the re-emission of nitrogen occur as a feedback related to changes in the total amount of nitrogen deposited from the atmosphere.

LAI sensitivity tests indicate that assigning oil palm plantation LAI from 3 to 6, as opposed to the 4.5 used in our simulations, has modest impacts on the changes in surface-atmosphere exchange over this region. Lower LAI reduces

isoprene emissions and dry deposition, and increases soil NOx emissions broadly across oil palm regions by ~5% compared to the Modern Palm simulation. The inverse is true of higher LAI values. The maximum changes in these processes are of order 10%.

## 3.2 Changes in Air Quality

Atmospheric composition over SEA is impacted by oil palm plantation expansion via the perturbations in surface-

atmosphere exchange discussed in Section 3.1. We focus here on how these changes connect to surface air quality in the region.





Formaldehyde (HCHO) is an oxidation product of isoprene, a toxic pollutant, and an $O_3$ precursor. Figure 6 shows the sensitivity of simulated HCHO to changes in the oil palm distribution. The largest increases in surface HCHO due to Modern Palm are in regions where surface fluxes of isoprene change the most, and are locally isolated due to the short atmospheric lifetime of HCHO (~hours). The largest relative increases in HCHO (up to 70%) are seen over northeastern Borneo, while

concentrations near the urban centers on the Malay Peninsula show increases greater than 50%. In terms of absolute values, the largest changes occur over Sumatra, with surface values increasing by as much as 2 ppbv. The increase over northeastern Borneo is lower, at around 1.4 ppbv. Across SEA, mean surface HCHO increases by 1.6%. HCHO sensitivities to Future Palm distributions share similar spatial characteristics to the Modern Palm scenario, with more pronounced changes over Kalimantan. Fractionally, the largest changes are in Sumatra, where surface concentrations of HCHO increase by up to a

factor of 1.8 compared to the No Palm baseline. Mean surface HCHO concentrations over SEA increase by 2.8% compared to the No Palm baseline. Absolute changes in surface HCHO are still quite high over Sumatra, above 2.5 ppbv in many regions.

Figure 7 shows that the simulated response of surface $NO_x$ to the oil palm expansion is very small. In principle, this response

is influenced by changes to deposition, soil $NO_x$ emissions, and isoprene fluxes. Given the modest difference in deposition and soil $NO_x$ emissions, the dominant impact is the elevated concentrations of isoprene. Additional isoprene leads to more conversion of NO to $NO_2$, and therefore increases the formation of $HNO_3$, leading to a net loss of $NO_x$. This effect is only apparent across the southern Malay Peninsula, a region with high surface $NO_x$ concentrations, due in large part to significant anthropogenic activity. These changes are typically less than 0.1 ppbv, on the order of 5% decreases. The Future oil palm

simulation shows similar decreases in the surface $NO_x$ response. These decreases are as large as 1 ppbv over Sumatra, a 5% drop. There is a decrease in $NO_x$ across Kalimantan on the order of ~0.5 ppbv related to the same chemistry. In reality, these changes may be dwarfed by the impact of anthropogenic emissions of $NO_x$ associated with production and processing facilities as well as oil palm fertilization; these changes are highly uncertain, and have not been included here.

The introduction of oil palm and the resulting increase in concentrations of isoprene can lead to changes in concentrations of ozone through VOC-$NO_x$ chemistry. At the same time, the increase in the deposition velocity of ozone leads to a shorter average lifetime, which decreases concentrations. In our modeled responses we see both of these signatures across SEA. Figure 8 shows that the surface ozone response to Modern Palm is most prominent over the southern Malay Peninsula (up to 4 ppbv), with changes over northeastern Borneo and Sumatra as well. Over the Malay Peninsula and Sumatra, a region not

sampled during OP3, surface ozone concentrations increase by up to 26% (3-4 ppbv) due to palm expansion. Ozone formation is enhanced in these regions, where additional isoprene emissions combine with $NO_x$ rich air near the major urban centers. Surface ozone increases in northeastern Borneo are on the order of 2 ppbv, located in the near vicinity of the oil palm plantations. These results differ spatially from Warwick et al. (2013), likely due to the substantially different land maps used for oil palm emissions of VOCs over Borneo. Hewitt et al. (2009) did not observe a change in surface $O_3$



concentrations due to oil palm at all over northeastern Borneo. Much of the discrepancy between our results and the Hewitt et al. (2009) observations can likely be explained by sampling and the different spatial resolution of the measurements and the model. The 0.5˚x0.666˚ grid box resolution used in this study is on the order of the entire study region for OP3.

Adding oil palm plantations usually increases the LAI (Figure 4), leading to an increased depositional velocity (Figure 5), which ultimately results in an increased sink of $O_3$. However, this is generally counteracted by the large increase in isoprene emissions. Our BVOC-only sensitivity simulation indicates that the changes in biogenic emissions are the dominant factor controlling the changes in $O_3$. The ratio of the changes in the Modern Palm simulation to those in the BVOC-only simulation is shown in Figure 9. Across most of the region, this ratio is nearly 1, indicating that the changes in soil $NO_x$ and

deposition have little impact on $O_3$. That said, over regions that undergo large forest to palm transitions and have relatively low background $O_3$ concentrations, such as Indonesian Borneo, up to 50% of the $O_3$ change is related to dry deposition and soil $NO_x$. A small decrease in the annual average $O_3$ concentrations (~10 pptv) is simulated over southwestern Borneo, where under low-NOx conditions, the additional isoprene from palm consumes $O_3$.

The changes in surface ozone are exacerbated in the Future Palm scenario (Figure 8). The relative surface concentrations over the Malay Peninsula increase by up to 4.5 ppbv (25%) by 2020 over the No Palm scenario. This is larger than the impact (<1 ppbv) estimated by Ashworth et al. (2012), likely due to their more modest estimate of palm expansion in 2020. $O_3$ concentrations in southern Kalimantan increase by up to 1 ppbv (~5%).

These changes have substantial impacts on local urban air quality in Kuala Lumpur and Singapore, which aim to adhere to the World Health Organization (WHO) (http://www.who.int/) guidelines for the daily maximum 8-hr average ozone not to exceed 50 ppbv. The number of days per year which exceed this standard in our simulation are shown in Figure 10 for both Singapore and Kuala Lumpur. The No Palm simulation suggests that surface $O_3$ exceeds the WHO standard for 35 days in Singapore and 23 days in Kuala Lumpur. We show how additional isoprene emissions associated with palm expansion

increases $O_3$ concentrations in these urban regions, exacerbating air quality issues. In particular, over Kuala Lumpur, Modern Palm is associated with 33 additional days of $O_3$ exceedance, increasing to 62 total days in the Future Palm scenario of 2020. The impacts over Singapore are more modest; nevertheless palm expansion is associated with 8 more days above the WHO guideline levels in 2020. From Figure 8, we observe that ozone air quality in Jakarta, another large urban center in the region, is relatively unaffected by oil palm expansion, due to the local transport patterns and the spatial distribution of the

plantations.

The changes in biogenic secondary organic aerosol (SOA) all track very closely with HCHO and isoprene, due to the rapid formation of SOA from biogenic precursors. These changes are shown in Figure 11. Relative increases in surface biogenic SOA concentrations due to Modern Palm expansion are highest in northeastern Borneo and the southern Malay Peninsula,





with increases larger than 60%, approximately 1 μg m$^{-3}$. Palm expansion in 2020 may lead to further substantial enhancements of SOA in the region, as high as 3.5 μg m$^{-3}$ and generally at least 1.5 μg m$^{-3}$ over regions with high oil palm density. Again, these values are larger than those in Ashworth et al. (2012), who employ a more modest oil palm expansion scenario. Palm expansion in the coming decade could lead to an average 5% increase in surface SOA across the region,

degrading visibility and enhancing air pollution exposure. Though most of these changes are local, they do stretch into protected nature preserves and dense urban regions.

The changes in air quality are not as sensitive to the choice of palm LAI (between 3 – 6), as compared to the changes in surface-atmosphere exchange processes. The magnitude of the changes in HCHO, O$_3$, and NO$_x$ are all on the order of 1%,

and not more than 5% relative to the Modern Palm scenario. The relative changes in SOA are also generally small, but are slightly higher (~±8%) over the southern Malay Peninsula, near Singapore.

## 4. Limitations of Observing Systems for Detecting the Impacts of Land Use Change on Air Quality

There are no long-term surface measurements of air quality in SEA to assess and validate our simulated impacts of oil palm expansion, beyond the snapshots provided by the OP3 campaign discussed above. However, a suite of space-based

instruments has been making global measurements during the peak of the palm expansion (2004 – Present). The rapid and extensive expansion of oil palm in SEA is arguably the most dramatic example of local land use change during the satellite era for atmospheric composition. We investigate whether the anticipated changes in air quality have been detectable from space over the last decade. We focus on the record of HCHO, O$_3$, NO$_2$, and Aerosol Optical Depth (AOD) observations, the suite of observable species that may have been impacted by oil palm development, and we use our model simulation to direct

this analysis.

All of the measurements analyzed here are part of the A-Train constellation of polar-orbiting satellites. As such they provide daily coverage in a sun synchronous orbit, with local overpasses ~10:00 and ~13:00. We use HCHO, NO$_2$, and O$_3$ measurements from the OMI instrument on board the NASA/Aura satellite, which has been operating since late 2004. The

satellite has a 14x24km footprint size. HCHO observations are from the NASA OMI HCHOv3 (OMHCHO) retrieval; filtering and quality control screening is described in Gonzalez Abad et al. (2015). These data have been used previously to successfully analyze large urban source regions and assess biogenic isoprene emissions (e.g. Zhu et al., 2014). We use the NO$_2$ retrievals from the DOMINOv2 retrieval product, as described in Boersma et al. (2011). These data have been used previously to assess emissions of NO$_2$ across Asia (Vinken et al., 2014). We use the NASA OMO3PR product (Bak et al.,

2015) that retrieves a vertical profile of ozone concentrations in 18 layers extending from the surface up to 0.3 hPa. There are alternative satellite measurements of all the above chemical species, but we focus on this suite of measurements that provide a consistent record during the peak palm expansion, aboard the same observing platform. We also explore Aerosol



Optical Depth (AOD) measurements from two MODIS instruments on board the NASA Terra and Aqua satellites. Our analysis uses the MODIS collection 6 product (Sayer et al., 2014).

All satellite data are filtered spatially to best capture the signal of palm expansion against the significant background of other sources. These include a large urban signature from the Kuala Lumpur, Singapore, and Jakarta megacities and substantial fire activity in both Sumatra and Indonesian Borneo. To address this, we focused our analysis on the remote northwest corner of Borneo, shown in the map of Figure 12. This contains the region where the airborne measurements were made during the OP3 campaign (Hewitt et al., 2010). Our model simulations suggest that this region has exhibited significant changes in air quality due to oil palm expansion, particularly in HCHO. Using Mientennen et al. (2010) as a base map, satellite measurements over potential palm plantation land types are identified within the "Palm" region, non-urban forest land-types in the "Forest" region, and ocean in the "Ocean" region. This classification allows for various time series analyses to be performed over what should be three distinct emission and depositional land types, and thus best isolate the impact of oil palm on local air quality.

Of all the atmospheric constituents observed, changes in HCHO concentrations should represent the strongest air quality perturbation due to oil palm expansion according to our model simulation (Figure 6). This is due to the strong local source of HCHO through oxidation of isoprene from oil palm, combined with the small urban sources in the region and the relatively constant background HCHO from methane oxidation. Background HCHO concentrations in our model simulation are on the order of $10^{16}$ molecules cm$^{-2}$. The background concentrations observed by the OMI instrument are also of the order $10^{16}$ molecules cm$^{-2}$. From our model simulations, the expected change in column concentration of HCHO due to Modern Palm expansion is of order $10^{15}$ molecules cm$^{-2}$. OMI is most sensitive to regions with very high HCHO concentrations ( $> 2 \times 10^{16}$ molecules cm$^{-2}$). Even at peak sensitivity, the instrument has an error of 30% per retrieval. This error compounds to more than 100% over areas with lower signals. The anticipated changes in HCHO due to oil palm expansion are therefore near the detection limits of the OMI instrument. Furthermore, the OMI sensor has been slowly degrading with time, causing a significant drop in data density since mid 2007 (Gonzalez Abad et al., 2015) due to an issue known as the "row anomaly". Instrument degradation is apparent in plots of annual average OMI HCHO, shown in Figure 13. Figure 13 suggests that HCHO concentrations have increased over SEA from 2005 through 2014, but that this is consistent with an overall increase in background HCHO that is not limited to SEA. This issue with instrument sensitivity, combined with a large decrease in the number of available observations makes it difficult to identify a significant trend in the HCHO satellite record. Figure 14 shows the monthly mean HCHO columns from OMI across all three selected land regions, as well as a LOWESS fit of the data with the shaded regions representing the 95% confidence interval obtained through a LOWESS block bootstrapping scheme (Cleveland, 1979). The measured HCHO over the forest and palm regions are both generally higher than over the ocean. The LOWESS analysis shows that HCHO concentrations are highest over the palm region, with a mean difference of ~0.6 x $10^{15}$ molecules cm$^{-2}$, slightly lower than but of similar magnitude to the difference expected from our model



simulations. This supports the results of our simulations, but also demonstrates how challenging it is to identify this signal from satellite observations. While the LOWESS analysis also suggests that the OMI HCHO column concentrations across the region increased over this time period, much of this may be driven by the row anomaly and we do not see evidence for significant palm plantation growth within our limited palm region selected in Figure 12.

There is large uncertainty with regard to the distribution and application of fertilizer on oil palm plantations (Mohd et al. 2015, Fowler et al. 2011) and industrial emissions of $NO_x$ associated with palm processing facilities (Hewitt et al., 2009). OMI $NO_2$ tropospheric retrievals are shown in Figure 15. Similar to the HCHO retrievals, both monthly mean columns and the LOWESS fit with 95% confidence intervals are shown. The $NO_2$ columns above the palm region are generally higher

relative to the forest region, but again there is no increasing difference over time. This is consistent with a lack of significant growth in fertilizer application or industrial emissions over the oil palm region selected in Figure 12. The constant palm-forest difference of $10^{14}$ molecules cm$^{-2}$ (~25%) agrees with our modeled analysis, which shows surface concentrations that differ by ~29% in this region, related to elevated anthropogenic $NO_x$ emissions. This suggests that we are not missing major palm-related sources of $NO_x$ emissions (fertilizer or industrial processing) in our simulation. The constant increase in

retrieved $NO_2$ concentrations over all three regions is consistent with Geddes et al. (2015b), who show a broad increasing trend in $NO_2$ across all of northern Borneo, possibly due to warmer surface temperatures, and transport from urban regions.

According to our simulations, the changes in other chemical species and aerosol will be most prominent over the large urban areas, or near areas of intense anthropogenic burning. The satellite-derived signal in tropospheric ozone from oil palm

development near Kuala Lumpur is not apparent against the background of urban development. Even though there are significant changes in the local ozone concentrations, too many confounding sources exist to identify the oil palm signal. Fires in SEA dominate the measured AOD, with an additional contribution from urban sources (Cohen and Lecoeur, 2015). Since the AOD measurement is a net observation of extinction from all aerosols at all altitudes, detecting changes in surface-level SOA is not straightforward. It is therefore challenging to identify the impact of oil palm expansion on air quality in

SEA with the current constellation of polar-orbiting satellites.

## 5. Conclusions

In this study, we simulate the impact of recent and projected (2020) oil palm expansion across SEA on air quality. We go beyond previous work by consistently treating the impact of land use change on a suite of land-atmosphere exchange processes relevant to atmospheric chemistry. Our simulations suggest that oil palm plantation expansion in the region has

had a significant impact on air quality. As oil palm expansion continues, the potential impact on surface $O_3$ concentrations is significant. The predicted ozone changes are largely due to increasing isoprene emissions. Locally however, increases in depositional velocities counteract these elevated emissions. If the oil palm crop expansion continues unabated, ozone



concentrations in urban regions could increase by up to 30% in 2020. Exposure to ozone is a significant cause of premature mortality, responsible for more than 200,000 deaths globally in 2013 (Forouzanfar et al., 2015). The increase in ozone attributable to oil palm plantations has the potential to bring many regions in SEA (including the dense urban areas of Singapore and Kuala Lumpur) further above WHO recommended threshold concentrations. The increase in particulate

matter due to biogenic secondary organic aerosol formation presents an additional health concern. Over Singapore, our results indicate that the addition of oil palm plantations contributes 9% of the target WHO recommended ozone concentrations, and 4% of the recommended 24 hour particulate matter concentrations. This is on the same order as the seasonal average predicted contribution of fires to the same particulate matter air quality targets (~8%) (Marlier et al., 2015b). This work illustrates that in trying to reach local air quality objectives, it is important to consider the impacts of local

land use change.

In this study we do not include the potential air quality impacts associated with local oil palm processing plants and prescribed burning of the fields. These are likely to lead to additional impacts on regional air quality, some of which have been described by Austin et al. (2015), Marlier et al. (2015a), and Hewitt et al. (2009). We also have limited constraints on

how depositional fluxes are altered by land use transitions, and the fertilization practices for oil palm. However our simulation of $NO_x$ and ozone deposition changes in the region appears to be broadly consistent with the measurements from the OP3 campaign.

Though the rapid oil palm expansion in SEA has led to substantial changes in the concentration of many atmospheric

species, including HCHO, $O_3$, and aerosols, many of these changes occur in areas with high fire and urban activity. Because of this, the signal of oil palm impacts on air quality is difficult to disentangle from the satellite record. This issue of strong confounding sources is compounded with issues of instrument sensitivity and degradation over time. This study suggests that future observing systems will require better sensitivities and stability to capture the impacts of land use change on air quality.

This work is an example of the impact of significant land use conversion on the regional scale. The growing pressures on the global food supply are likely to lead to further land conversions to support agricultural activity in the coming decades. Future atmospheric composition will respond to these changes, with implications for air quality and climate, and will remain important to monitor and understand.

**Acknowledgements**

This work was supported by NSF (AGS-1238109).



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


**Figures**

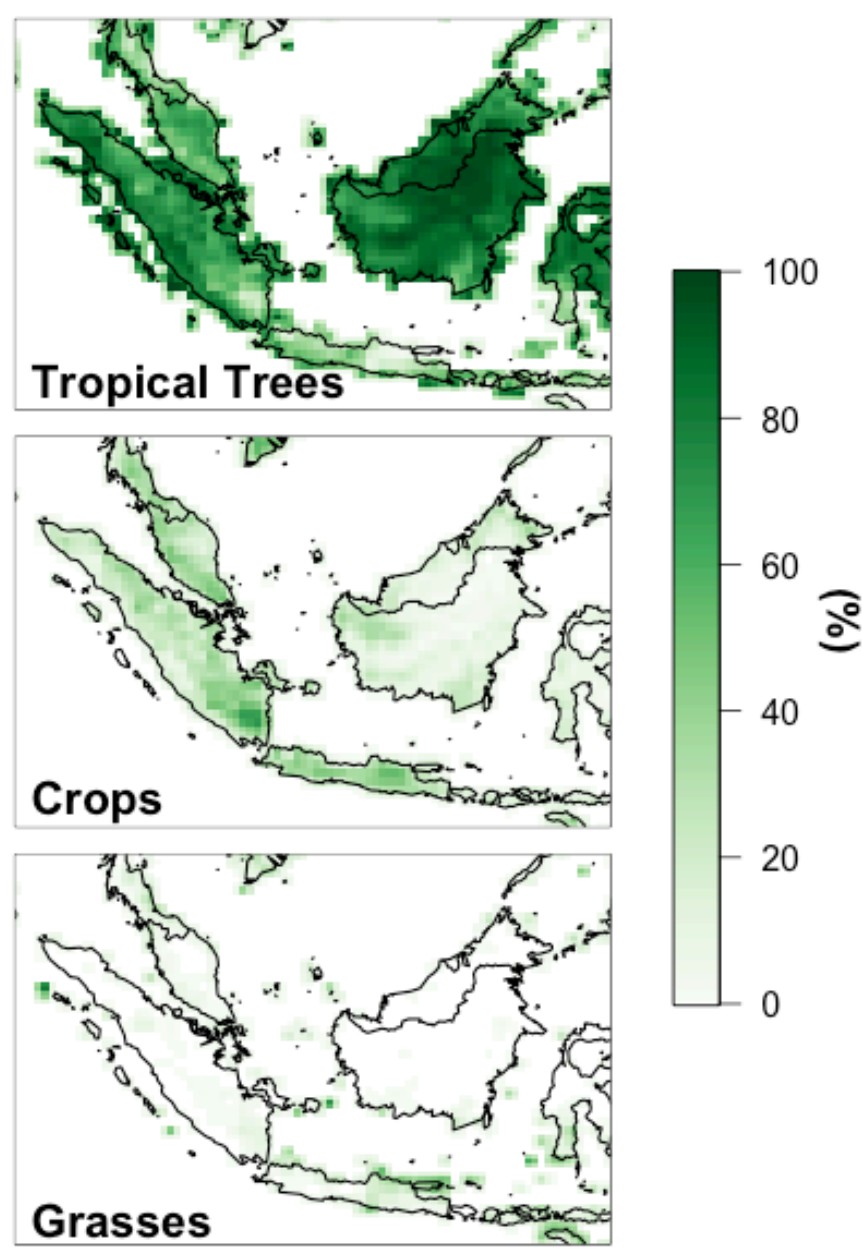

**Figure 1: Percentage of vegetated area occupied by dominant vegetation classes in the No Palm scenario at the native 0.23˚x0.31˚ resolution.**





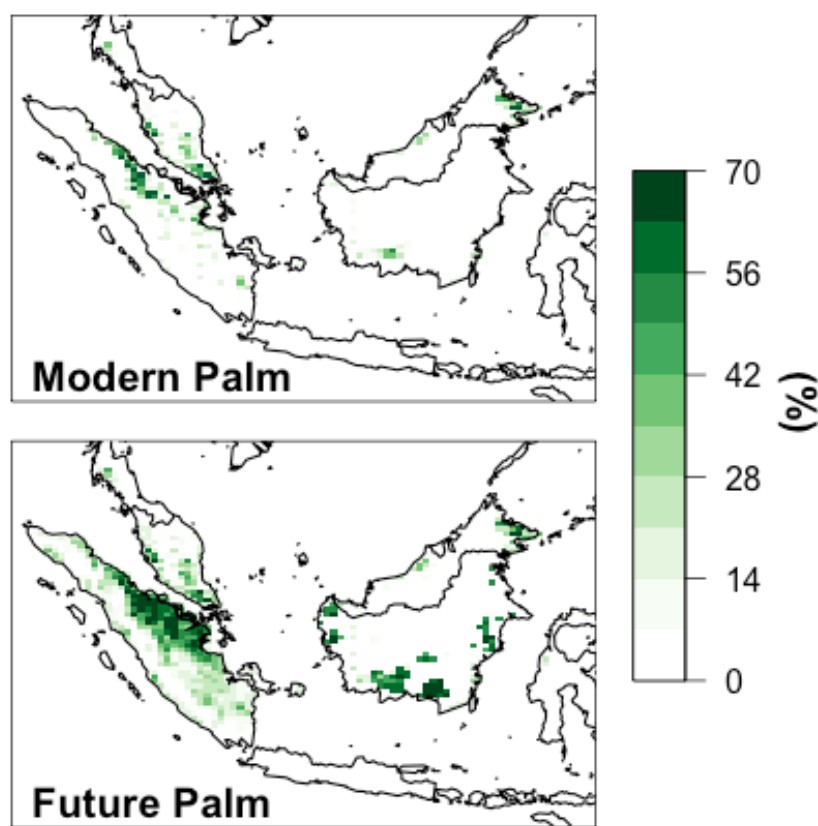

**Figure 2: Percentage of vegetated area occupied by oil palm plantations for Modern (2010) and Future (2020) scenarios. Note that estimates for palm planatation increases from 2010 to 2020 are only avaialble for Sumatra and Kalimantan; palm plantation coverage in other regions are assumed constant.**



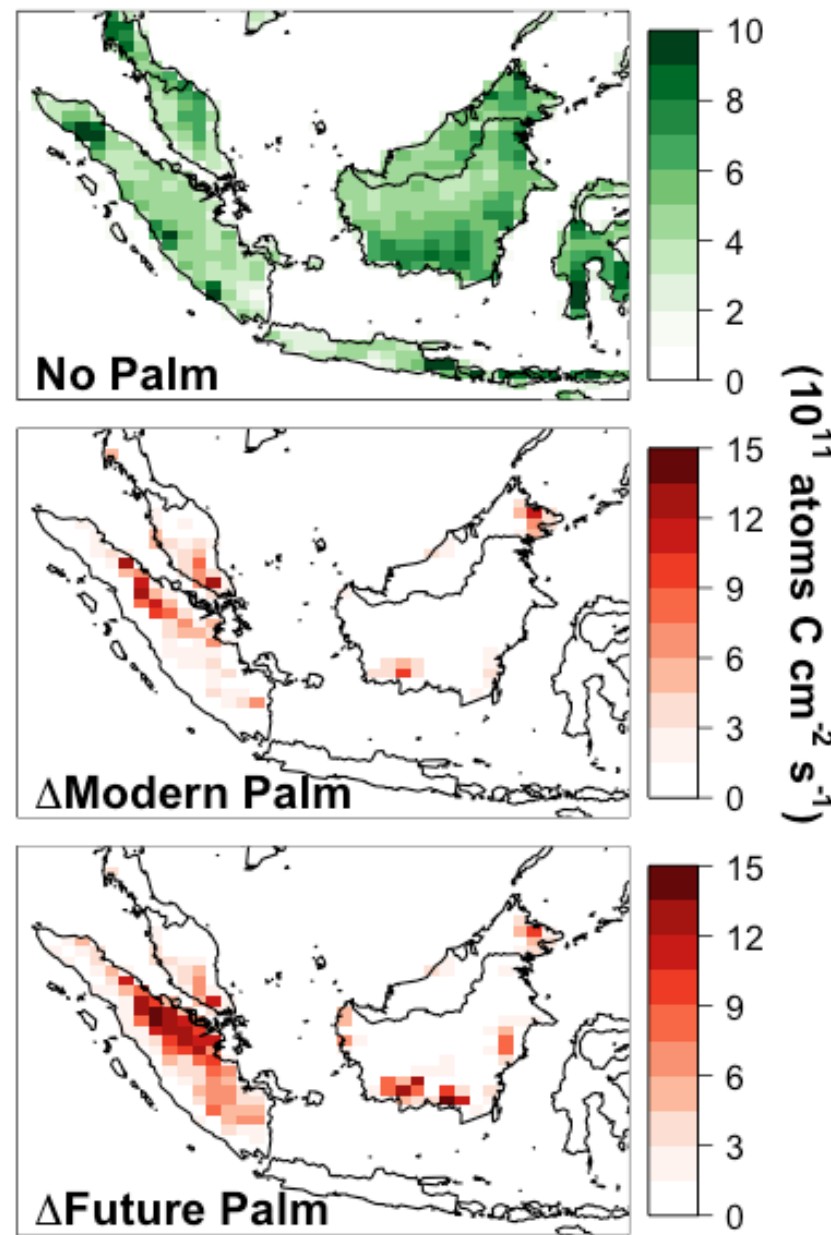

**Figure 3: Annual mean simulated isoprene emissions over SEA (top) and the change due to Modern (middle) and Future (bottom) oil palm expansion.**


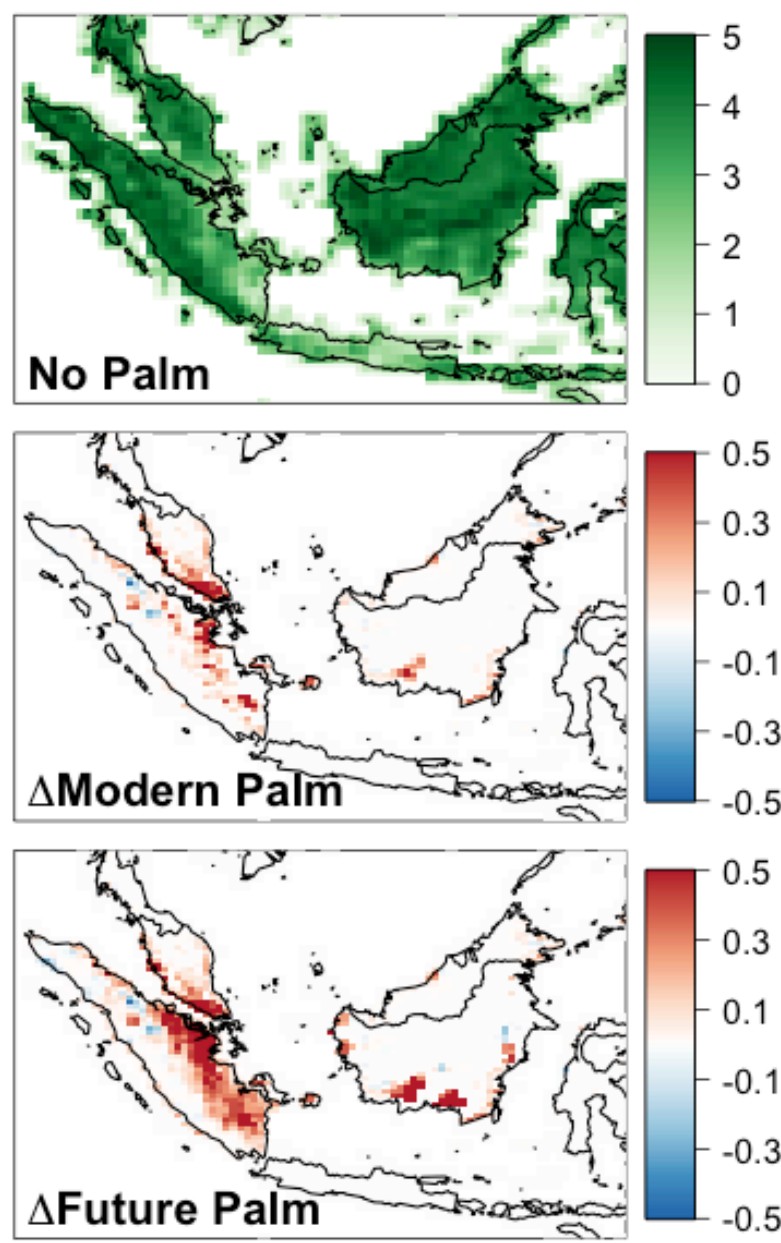

**Figure 4: Annual average leaf area index (LAI) over SEA (top) and the change due to Modern (middle) and Future (bottom) palm expansion.**



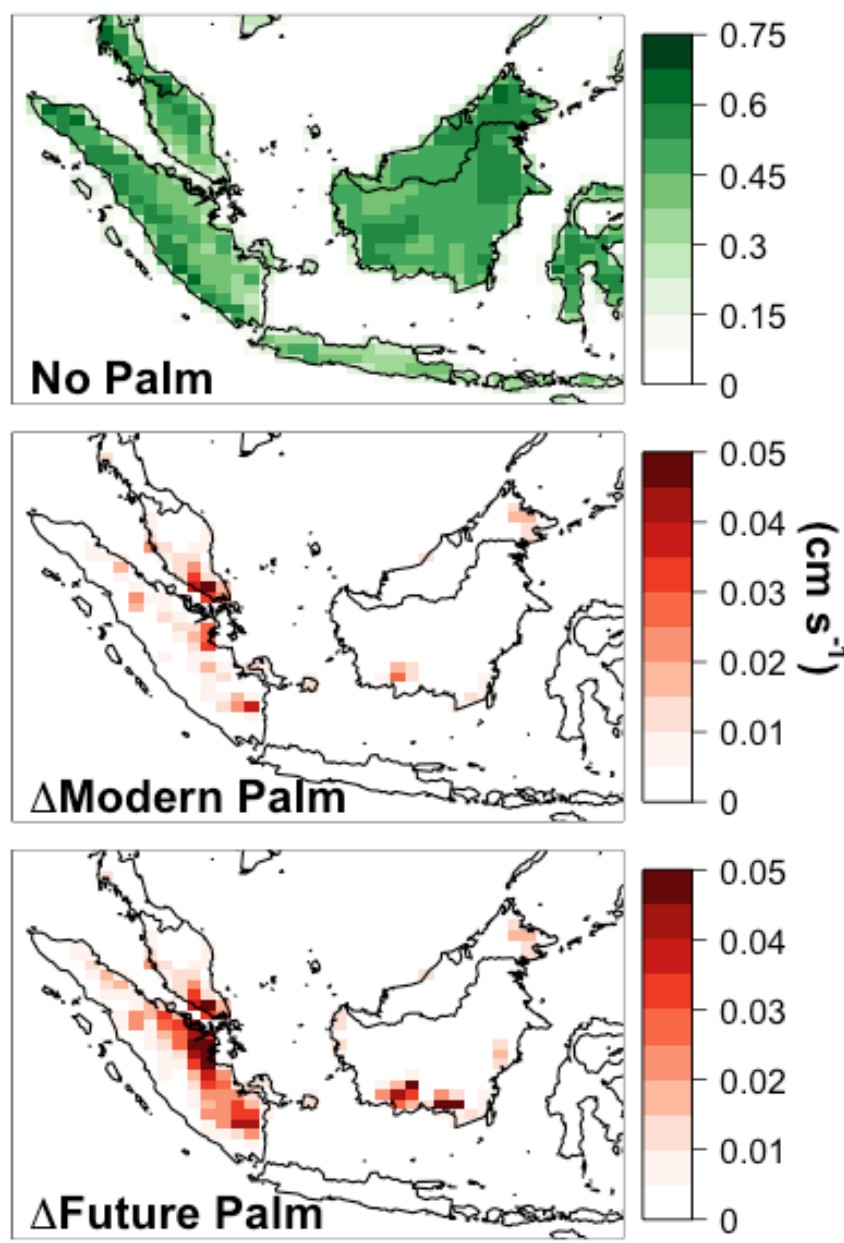

**Figure 5: Annual average ozone dry deposition velocity over SEA (top) and the change due to Modern (middle) and Future (bottom) palm expansion.**





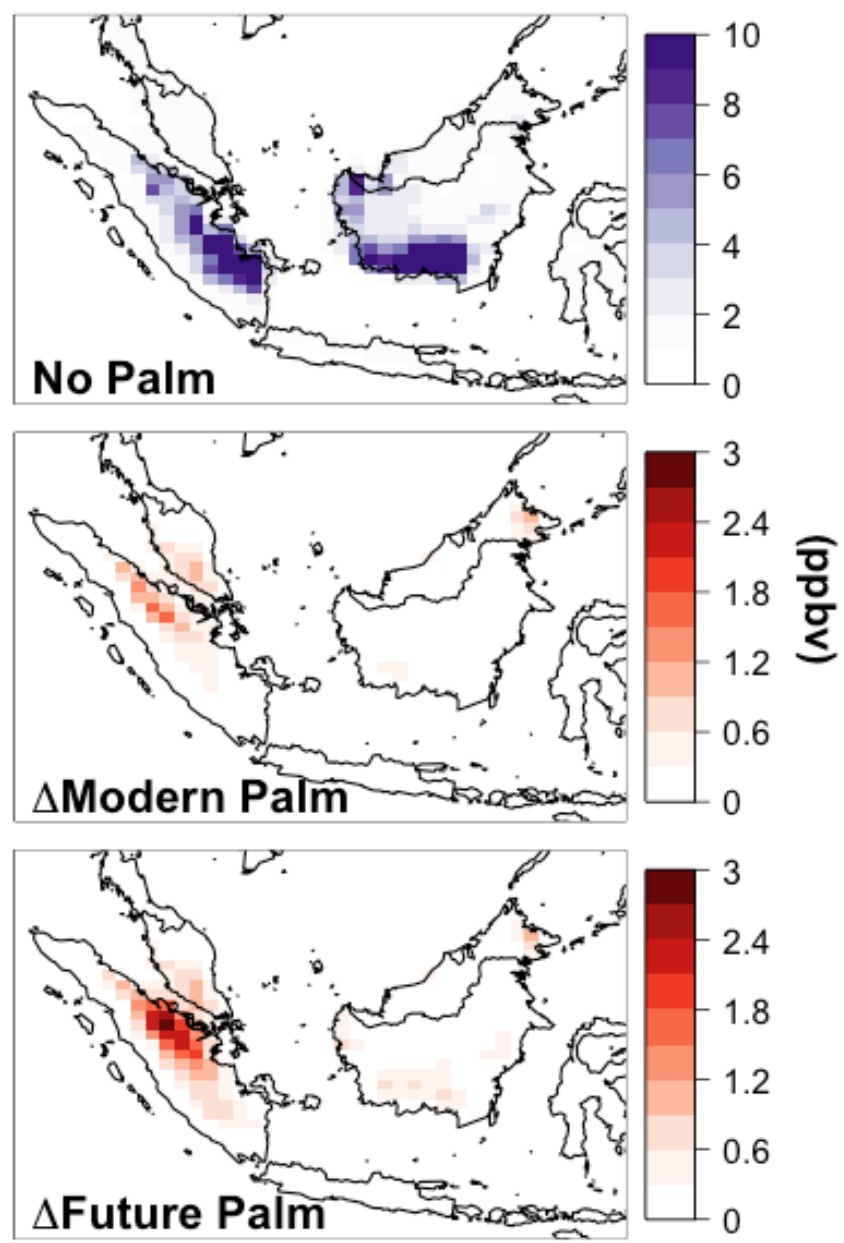

**Figure 6: Annual average surface formaldehyde (HCHO) concentrations over SEA (top) and the change due to Modern (middle) and Future (bottom) palm expansion.**



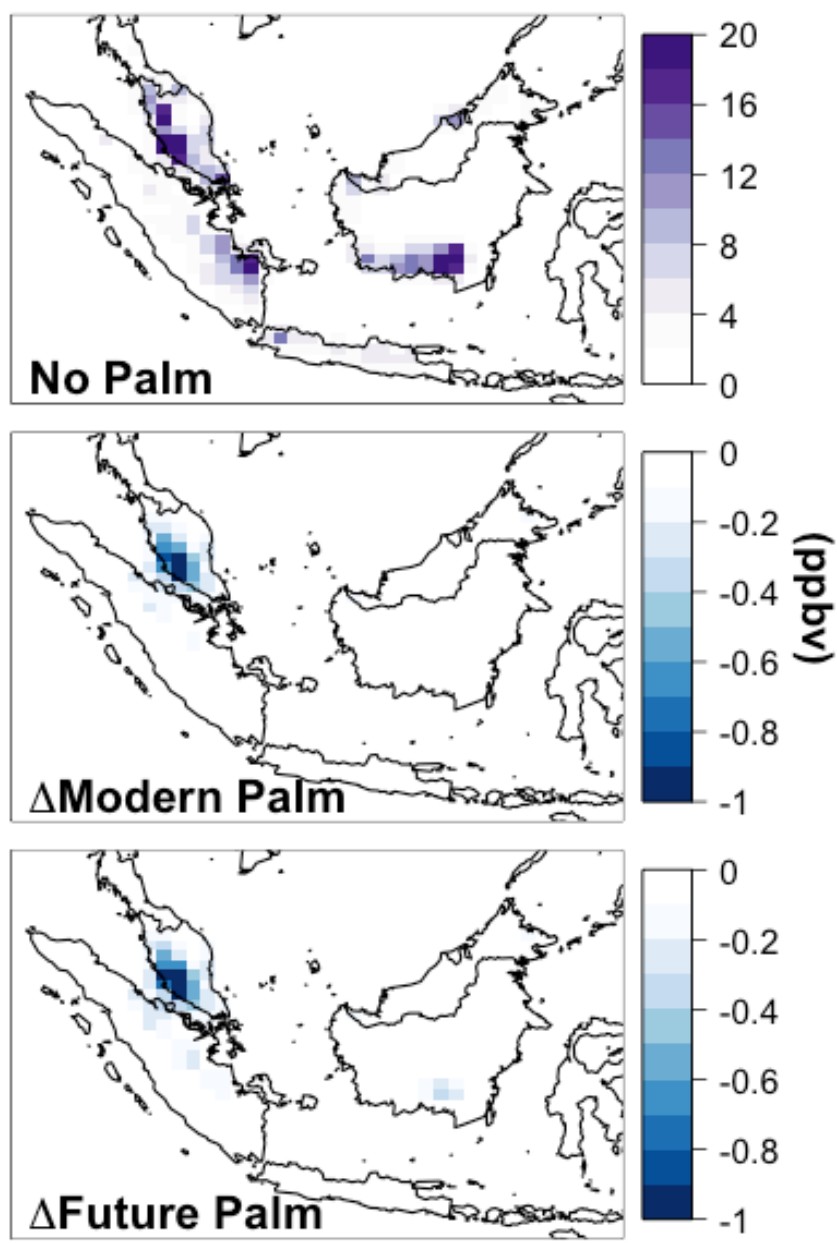

**Figure 7: Annual average surface nitrogen oxides (NO$_x$) concentrations over SEA (top) and the change due to Modern (middle) and Future (bottom) palm expansion.**



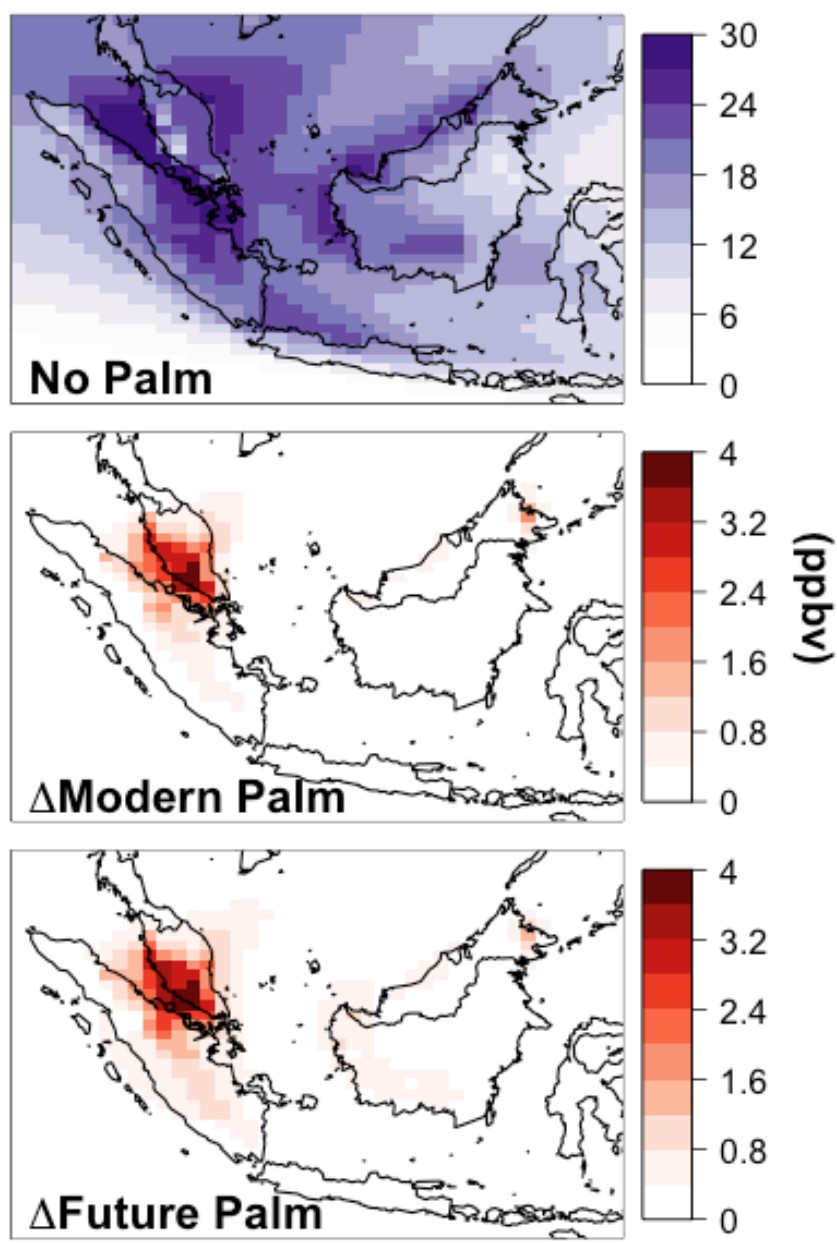

**Figure 8: Annual average surface ozone concentrations over SEA (top) and the change due to Modern (middle) and Future (bottom) palm expansion.**





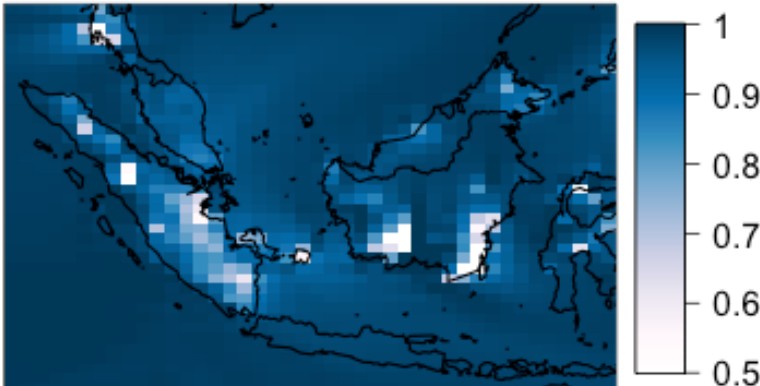

**Figure 9: The ratio of the changes in ozone concentrations in the Modern Palm scenario compared to the changes in ozone in the BVOC-only simulation. Regions less than 1 show where increasing deposition velocities over palm plantations counteract some of the ozone increases driven by increasing isoprene emissions.**

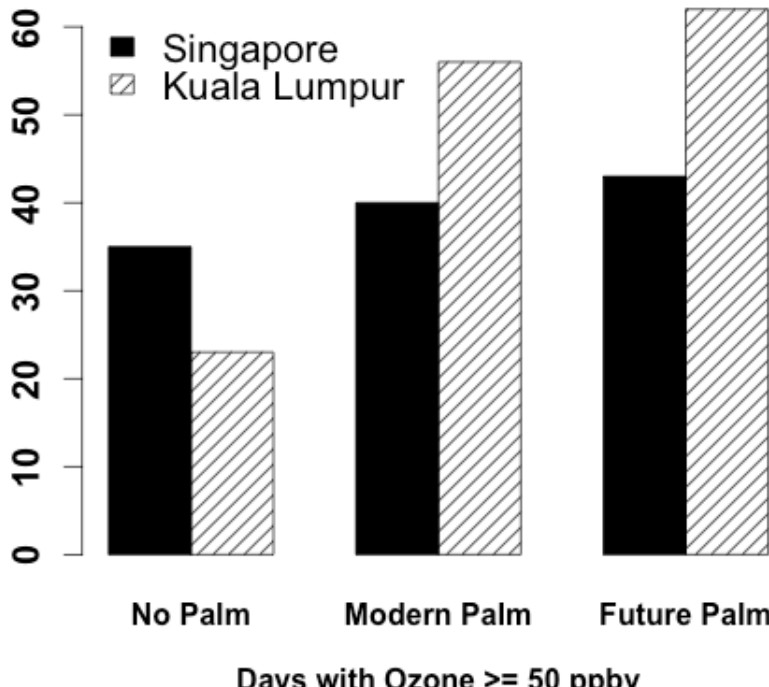

**Figure 10: The number of days with simulated daily maximum 8-hour average surface ozone concentrations exceeding 50 ppbv over Singapore and Kuala Lumpur.**





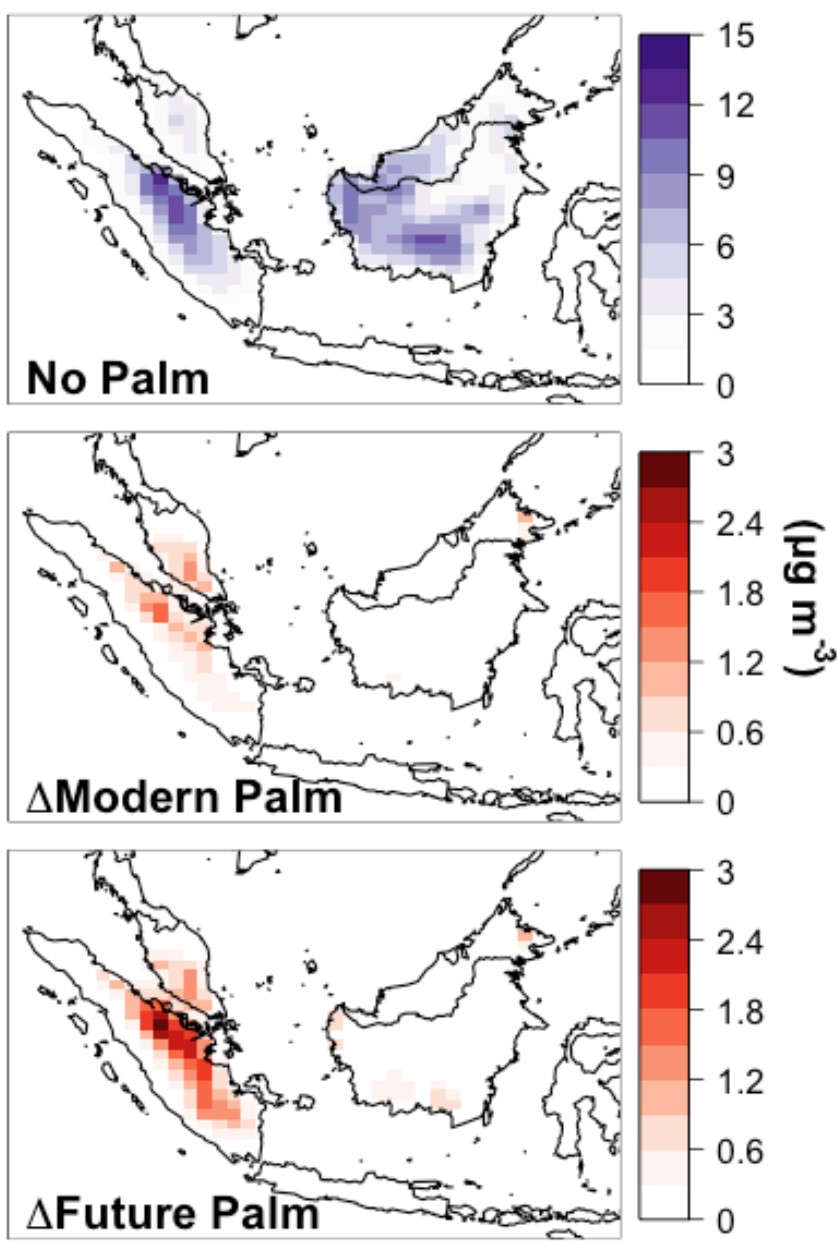

**Figure 11: Annual average surface biogenic secondary organic aerosol (SOA) concentrations over SEA (top) and the change due to Modern (middle) and Future (bottom) palm expansion.**





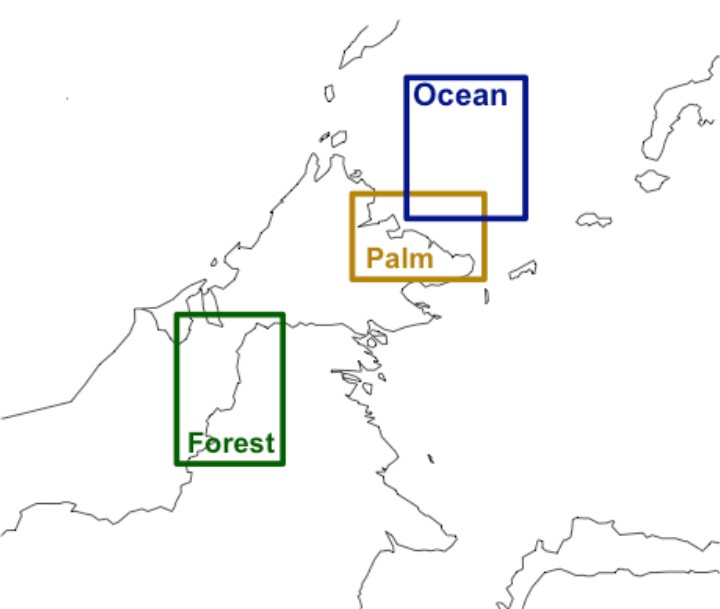

**Figure 12: The three regions in northeastern Borneo used for the spatial filtering of the satellite data.**



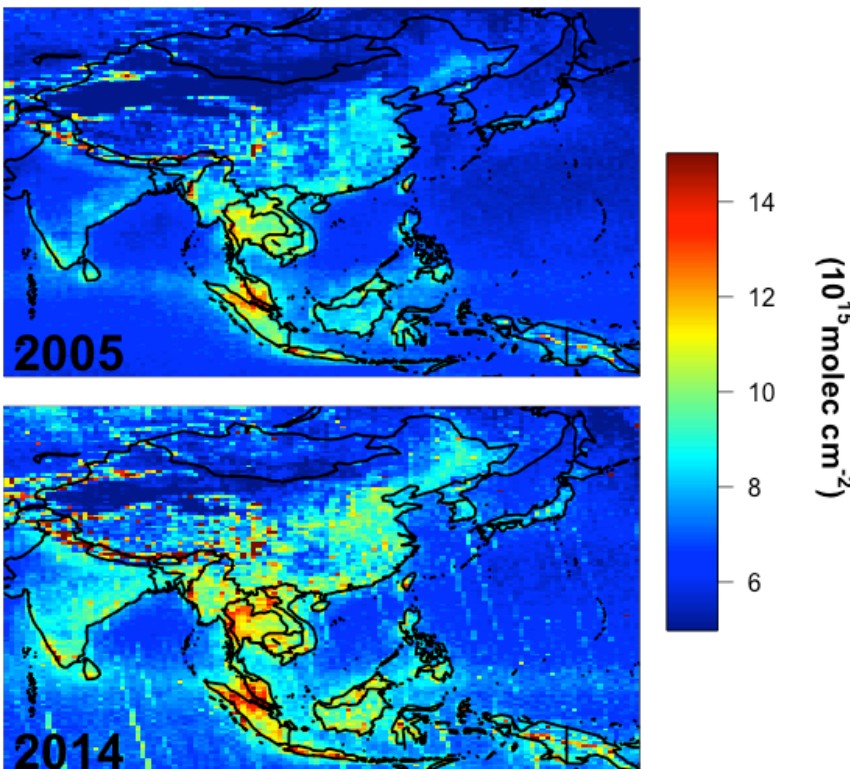

**Figure 13: Annual average HCHO columns measured by the OMI instrument in 2005 and 2014.**





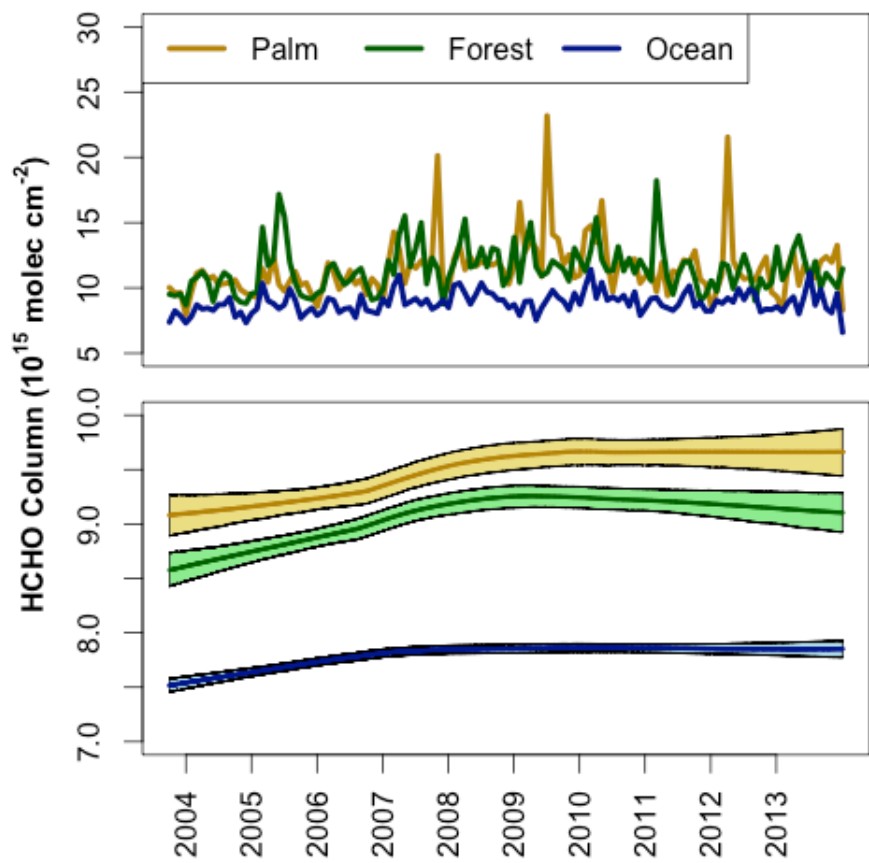

**Figure 14: A monthly mean and LOWESS timeseries of HCHO from OMI in the three separate regions of northern Borneo (see Figure 12). The LOWESS shaded regions represent the 95% confidence interval.**





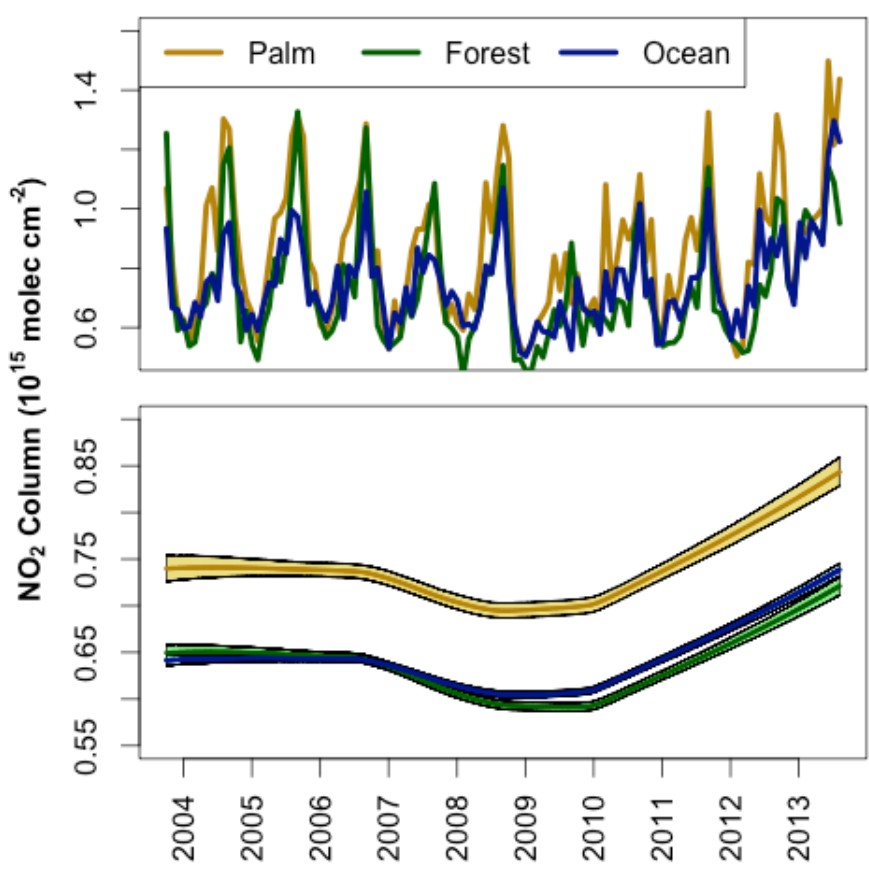

**Figure 15: A monthly mean and LOWESS timeseries of NO₂ from OMI in the three separate regions of northern Borneo (see Figure 12). The LOWESS shaded regions represent the 95% confidence interval.**