# Peer review of "Impacts of Current and Projected Oil Palm Plantation Expansion on Air Quality Over Southeast Asia"

_Atmospheric Chemistry and Physics, 2016_

## Referee Comment (RC1) · Anonymous Referee #1 · 25 Mar 2016

Silva et al. perform a model investigation into the impacts of a specific type of land cover change onto regional air quality in southeast Asia. They test the sensitivity of surface O3 and aerosol AOD to the historical expansion of palm oil plantations in Kalimantan and Sumatra and to a "business-as-usual" projection of plantation expansion to 2020. Increases in ozone deposition velocity driven by the plantation expansion into primary vegetation counterbalance somewhat the enhanced ozone from the oil palm isoprene emissions. Still, the model results suggest that poor air quality events are increased in some nearby urban centers from the expansion of the plantations and could be further increased in the near future. Attempts to support the results of the simulations with satellite-derived data were inconclusive given the difficulty in isolating the relatively small palm oil impact on regional chemistry.

I found this to be a well-designed study that reached interesting conclusions regarding

terrestrial impacts on urban air quality. These types of impacts are often overlooked but this study is a good example for how land cover changes can be important. The manuscript is also well written in my view with an appropriate amount of detail about the chemistry and model setup included, and clearly presented figures. I recommend publication after minor revisions that are listed below.

General comments: 1. I wonder if anything can be said about the credibility of the year 2020 projections since we are only four years away from 2020. There are some data available on forest area for 2015 from the FAO (http://www.fao.org/3/a-i4808e.pdf) which show about the same rate of forest loss in Indonesia between 2010-2015 as in 2005-2010, although not a perfect proxy for palm oil expansion in the two specific regions considered in this study.

2. It would be helpful to include the locations and names of the major urban areas referred to in the text in one of the early figures, possibly figure 1a, especially given that their relative locations are important for palm oil plantation impacts.

Minor comments:

Pg 4, Line 3: Can you comment on how the high fire emissions and dry conditions associated with the beginning of an el nino event in late 2006 may or may not impact your annual average results?

Pg 5, Line 4: I think the "native forest" here refers not to a new PFT but to broadleaf evergreen trees as mentioned later in the sentence, this could be made more clear so the reader understands exactly what is being changed. The need to reduce the isoprene emission factor by 4 times over the default seems like it could be a major issue for global studies given that this forest PFT is dominant in the region, topic for another study maybe.

Pg 5, Line 9: Delete "that"

Pg 6, Lines 12-14: Does the palm oil PFT always replace the tropical forest PFT or does

it replace all PFTs in a grid cell, including crops and grasses, based on the original fraction of the PFTs? It is suggested later on in the text that other land cover types besides forest are being converted to palm oil (Pg 8, Line 2) but it would be helpful to have this made clear on Pg 6.

Pg 8, Line 26-27: Is the 10% mentioned here also relative to the Modern Palm simulation?

Pg 12, Lines 10-13: Does the overlap in the "Palm" and "Ocean" filtering regions dilute the palm oil signal in the results shown in figures 14 and 15?

Pg 13, Lines 18-19: This sentence was a bit unclear to me and I think the rest of the paragraph stands well enough on its own such that this sentence could probably be deleted.

Pg 14, Line 1: The 30% increase in ozone – is this relative to the no-palm world, or to 2010, and is this the palm oil plantation contribution alone or does this include increases in other sources of ozone?

---

## Referee Comment (RC2) · Anonymous Referee #2 · 11 Apr 2016

General comments

The authors present a study of land conversion to oil palm plantations in SE Asia, although this is limited to Indonesian Borneo and Sumatra. They use the GEOS-Chem atmospheric chemistry transport model to investigate how changes in land cover affect emissions of volatile organic compounds (specifically isoprene) and the impacts this has on atmospheric composition in the region. They find that increasing the area of oil palm plantations increases isoprene emissions and hence ozone and aerosol concentrations over most of the region.

While this is clearly a topic of interest, and one that is central to the scope of ACP, the research presented here is not sufficiently novel in my opinion to warrant publication at this time.

Specific comments

The main issues that I have with this manuscript are:

1. Novelty:

I do not feel that the work presented here is sufficiently novel in scope or methodology to mark an advance on previous work. Other than the use of different land use change maps the simulations do not differ from previous studies nor does the analysis of the model output extend beyond what has been considered before.

The authors introduce a new oil palm-specific plant functional type (PFT) into GEOS-Chem, adapting isoprene emission factor and LAI of the existing tropical broadleaf evergreen tree PFT using data from the OP3 field campaign. This is the exact approach taken by Ashworth et al., 2012 and Warwick et al., 2013, which built on earlier investigations of LUC (although not specifically due to oil palm) by e.g. Wiedinmyer et al., 2006; Lathiere et al., 2006; Ganzeveld et al., 2010.

The authors demonstrate that changes in oil palm distribution alter isoprene emissions (and hence concentrations) and thence concentrations of O3 and SOA. This is not a new finding (Ashworth et al., 2012; Warwick et al., 2013, and many other studies showing that different land cover affects atmospheric composition via changes in biogenic VOC emissions, e.g. Guenther et al., 2006; Arneth et al., 2011; and those listed above). The authors' results differ only in terms of distribution and scale.

Including changes in NOx emissions concomitant to the changes in land cover is not new, and in fact the study here does not go as far as Ashworth et al., 2012 who included sensitivity tests with and without NOx emissions associated with processing, nor Warwick et al., 2013 who included changes in soil NOx emissions associated with periodic fertilization of oil palm plantations.

Changes in O3 deposition have also been included in many previous studies of LUC (e.g. Ganzeveld et al., 2010; Ashworth et al., 2012).

[Figure]

Furthermore, since the publication of the OP3 data that the authors cite here, further data have been presented reporting high emissions of other VOC from oil palm (e.g. methyl chavicol (estragole) and toluene, Mizstal et al., 2010; 2011; 2015).

Higher than expected deposition of a number of other compounds has also been reported (e.g. Karl et al., 2009; Nguyen et al., 2014) and yet this does not appear to have been considered by the authors who refer to the reactivity of ozone as an additional reason to focus on its deposition. I would also be interested to know if GEOS-Chem partitions dry deposition between stomatal and non-stomatal routes in line with e.g. Fares et al., 2012; 2013; 2014; Simpson et al., 2012.

2. Methodology:

The spatial resolution of the model (0.5x0.66667deg) is too coarse for studying air quality (see e.q. Gego et al., 2005; Varghese et al., 2011; Schaap et al., 2015). In my view, this is a study of impacts on atmospheric composition rather than air quality and should be so described (i.e. in the title and text). Further, the authors only demonstrate how the projected changes in O3 concentrations relate to recognized WHO air quality standards although they discuss changes in formaldehyde (not a regulatory air pollutant), NOx and SOA as well. And yet, premature mortality and morbidity associated with particulate matter is almost an order of magnitude higher than for O3. I also find the choice of metric odd; the number of exceedance days is a threshold metric (i.e. a consideration of "extreme" conditions) which is likely to be poorly represented by a coarse resolution model.

The temporal resolution of the quoted changes in atmospheric composition is also not sufficient for air quality assessments. While annual limits are given for some pollutants (although mostly in terms of accumulated exposure), daily 8-hour and peak 1-hour exposure is the more normal metric considered. Presenting changes in annual average concentrations is therefore inappropriate in the context of air quality.

It appears that GEOS-Chem was driven with meteorology for a single year (2006).

The authors report that there was no substantial difference in projected changes in atmospheric composition between seasons. This is in contrast to the findings reported by Ashworth et al., 2012, and seems odd given that SE Asia is a monsoon-influenced region. That, plus the high level of fires reported for 2006, suggest that it may not have been a "typical" or representative year. Did the authors give any consideration to the inter-annual variability of their findings?

A future scenario set in 2020 seems rather limited in scope given that it is now 2016. It would have been interesting to assess how the LUC might combine with future changes in climate and air quality in the region with a longer-term scenario.

3. Other:

The analysis is limited with changes in atmospheric composition given almost entirely in terms of changes in annual averages. On the whole, presentation of results is limited to a series of virtually identical figures. As most of the changes are spatially similar there seem an unnecessary number of figures. They do highlight the issue of model resolution quite clearly. Pugh et al., 2013 identified SE Asia as a region in which model spatial resolution is particularly important for atmospheric chemistry modeling which also appears not to have been considered by the authors.

The choice of color scale for Figure 9 is poor. It is virtually impossible to make out the outline of the islands when this is printed out. Using white for a ratio of unity would seem a more sensible way to show the limited extent of the impact.

Isoprene emissions are not usually given in units of atoms C cm-2 s-1 in the context of a regional modeling.

---

## Author Comment (AC2) · 10 Jun 2016

We thank the referee for their consideration of our manuscript. Below are our responses to each of the comments, including the proposed changes to our revised manuscript.

**General Comments**
The authors present a study of land conversion to oil palm plantations in SE Asia, although this is limited to Indonesian Borneo and Sumatra. They use the GEOS-Chem atmospheric chemistry transport model to investigate how changes in land cover affect emissions of volatile organic compounds (specifically isoprene) and the impacts this has on atmospheric composition in the region. They find that increasing the area of oil palm plantations increases isoprene emissions and hence ozone and aerosol concentrations over most of the region.
While this is clearly a topic of interest, and one that is central to the scope of ACP, the research presented here is not sufficiently novel in my opinion to warrant publication at this time.

*We appreciate the reviewer's in depth analysis and comments on our work, however we disagree with several key points made by the reviewer. We note these throughout the responses.*

*We would also like to clarify the reviewer's characterization that that the work is "limited to Indonesian Borneo and Sumatra". We include the oil palm distribution across all of Malaysia as well, thus capturing >90% of all global oil palm agriculture. For the future scenario sensitivity, we only change the distribution of oil palm over Sumatra and Kalimantan due to a lack of information on the future distribution of palm throughout Malaysia. We believe that this is an acceptable sensitivity test because, as stated in our manuscript, the majority of land across both Indonesia and Malaysia available for oil palm plantation expansion is on Sumatra and Kalimantan.*

**Novelty**
I do not feel that the work presented here is sufficiently novel in scope or methodology to mark an advance on previous work. Other than the use of different land use change maps the simulations do not differ from previous studies nor does the analysis of the model output extend beyond what has been considered before.

*We have updated the text to further emphasize the novelty of the work (described in the detailed points below).*
*We further note that the reviewer did not discuss the novel satellite analysis presented in Section 4, wherein we describe how and why current observing systems are not capable of detecting the air quality impacts of the massive land use change signature of oil palm. Our conclusions are critically important for implementation of observing systems of the future. If we cannot observe the impacts of the massive change in biogenic emissions associated with oil palm plantations based on current satellite measurement capabilities, we are likely to struggle to detect the impacts of any forest to agriculture land use conversions in the future.*

The authors introduce a new oil palm-specific plant functional type (PFT) into GEOS- Chem, adapting isoprene emission factor and LAI of the existing tropical broadleaf evergreen tree PFT using data from the OP3 field campaign. This is the exact approach taken by Ashworth et al., 2012 and Warwick et al., 2013, which built on earlier investigations of LUC (although not specifically due to oil palm) by e.g. Wiedinmyer et al., 2006; Lathiere et al., 2006; Ganzeveld et al., 2010.

*Our implementation of the oil palm-specific plant functional type differs from both Ashworth et al. (2012) and Warwick et al. (2013) in several key ways:*

*Ashworth et al. (2012) scaled the isoprene emissions of only broadleaf evergreen trees within a given grid box based on the fraction of oil palm expected within that grid box. The scaling factor used was a ratio of measured emission factors for oil palm and broadleaf evergreen trees: 50/35. For this work, we reduce the fraction of all vegetation in a grid box proportionally, not just broadleaf evergreen trees. This accounts for the fact that oil palm is not only farmed in regions where the natural rainforest has been removed. We additionally directly calculate the isoprene emissions through the MEGANv2.1 algorithm using the measured basal isoprene emission factors from OP3. By using the MEGAN algorithm in conjunction with the measured emission factors, we are able to more robustly estimate oil palm emissions outside of the timeframe of the OP3 field campaign (accounting for seasonal temperature differences, PAR, LAI, etc.) This was not explicitly accounted for in Ashworth et al. (2012) or Warwick et al. (2013).*

*Warwick et al. (2013) explored a future scenario where the entire island of Borneo was covered with oil palm vegetation, and replaced the MEGAN emissions algorithm with emissions measured during the OP3 field campaign. They acknowledge that this is "obviously an extreme situation", but that is useful for exploring a certain air quality trajectory. We use a more realistic land map for both modern and near-term future oil palm distributions, including the distribution of oil palm on Sumatra and the Malay Peninsula.*

*We clarify some of these differences in our manuscript on P2 Lines 31-34. And P5 L9- 11*

The authors demonstrate that changes in oil palm distribution alter isoprene emissions (and hence concentrations) and thence concentrations of $O_3$ and SOA. This is not a new finding (Ashworth et al., 2012; Warwick et al., 2013, and many other studies showing that different land cover affects atmospheric composition via changes in bio- genic VOC emissions, e.g. Guenther et al., 2006; Arneth et al., 2011; and those listed above). The authors' results differ only in terms of distribution and scale.

*We agree that the finding that changes in isoprene emissions can change $O_3$ and SOA concentrations is not unique to this study. We argue that the differences that we model in terms of distribution, scale, and magnitude are important enough to consider this*

*work novel. In particular, our simulations explore the integrated effects of changing emissions AND deposition and are at higher resolution (0.5x0.67) than Ashworth et al. (2012) and Warwick et al. (2013). Previous studies also do not explore the impact of oil palm on densely populated regions, such as the Malay Peninsula, where we see the largest surface O3 increases.*

Including changes in NOx emissions concomitant to the changes in land cover is not new, and in fact the study here does not go as far as Ashworth et al., 2012 who included sensitivity tests with and without NOx emissions associated with processing, nor Warwick et al., 2013 who included changes in soil NOx emissions associated with periodic fertilization of oil palm plantations.

*We agree with the reviewer: the treatment of NOx emissions is not a novel aspect of our study. Given the uncertainties associated with fertilization (as discussed for the OP3 field campaign by Fowler et al. (2011)) and processing emissions, changes in $NO_x$ are not a focus of this study. More information on additional $NO_x$ sources from fertilization or processing is needed to realistically explore this. However, we note that our satellite analysis in Section 4 confirms that we are not missing major palm-related sources of $NO_x$ in our simulation, suggesting that these sources may indeed be modest.*

Changes in O3 deposition have also been included in many previous studies of LUC (e.g. Ganzeveld et al., 2010; Ashworth et al., 2012).

*We agree that changes in $O_3$ deposition have been included in studies of land use change before. However, the simultaneous exploration of both emissions and deposition changes related to oil palm had not been studied before. Warwick et al (2013) and Ashworth et al. (2012) both explored separate sensitivity studies to show the potential influence of deposition. Warwick et al (2013) doubled all deposition velocities, to test their model sensitivity. Ashworth et al. (2012) perform several short sensitivity studies of deposition, and conclude that the changes in deposition are likely to increase $O_3$ concentrations. This is at odds with our work, where we show the opposite through a complete simulation of the depositional changes.*

*The Ashworth et al. (2012) result is likely due to the fact that they used a higher biomass density for the natural vegetation than for oil palm plantations, and only allowed oil palm to replace the tropical broadleaf evergreen trees. This reduction in biomass density led to a reduction in deposition. Our work uses a dry deposition scheme (Wesely and Hicks, 2000) that does not use biomass density as a parameter, but instead uses LAI. This is likely a more realistic parameter due to the heavy dependence on stomatal and cuticular deposition in heavily forested regions. The observationally-derived land cover inputs used in this work show that oil palm plantations have higher LAI than most of the natural rainforest and other land types across Southeast Asia. This leads to the increased deposition in our work.*

*We added text to P6 L4-8 and P6 L27-33*

Furthermore, since the publication of the OP3 data that the authors cite here, further data have been presented reporting high emissions of other VOC from oil palm (e.g. methyl chavicol (estragole) and toluene, Mizstal et al., 2010; 2011; 2015).

*The reviewer correctly points out that the oil palm plantations have higher emissions of estragole and toluene, however their chemical transformations (and thus impact on $O_3$ and SOA) are not well constrained, and Guenther et al. (2012) indicate that emissions of these species are relatively unimportant, particularly as compared to isoprene. As a result, we do not include these species in our analysis and do not believe that this omission significantly impacts our results. We add a sentence to P5 L18-20 discussing this.*

Higher than expected deposition of a number of other compounds has also been reported (e.g. Karl et al., 2009; Nguyen et al., 2014) and yet this does not appear to have been considered by the authors who refer to the reactivity of ozone as an additional reason to focus on its deposition. I would also be interested to know if GEOS-Chem partitions dry deposition between stomatal and non-stomatal routes in line with e.g. Fares et al., 2012; 2013; 2014; Simpson et al., 2012.

*The changes in land cover do change the deposition of other compounds in GEOS-Chem. We focus on $O_3$ due to the significance of $O_3$ on regional air quality, and because its reactivity makes it a good candidate for describing the changes in deposition related to oil palm expansion. The magnitude of changes in deposition for all other species is nearly always less than 10%, and never more than 15%. The dry deposition module in GEOS-Chem does consider deposition through both stomatal and non-stomatal pathways, as outlined in Wesely and Hicks (2000), but only calculates one net surface sink.*

**Methodology**
The spatial resolution of the model (0.5x0.66667deg) is too coarse for studying air quality (see e.q. Gego et al., 2005; Varghese et al., 2011; Schaap et al., 2015). In my view, this is a study of impacts on atmospheric composition rather than air quality and should be so described (i.e. in the title and text). Further, the authors only demonstrate how the projected changes in O3 concentrations relate to recognized WHO air quality standards although they discuss changes in formaldehyde (not a regulatory air pollutant), NOx and SOA as well. And yet, premature mortality and morbidity associated with particulate matter is almost an order of magnitude higher than for O3. I also find the choice of metric odd; the number of exceedance days is a threshold metric (i.e. a consideration of "extreme" conditions) which is likely to be poorly represented by a coarse resolution model.

*This model resolution has commonly been used to study air quality in Southeast Asia, including: Kim et al. (2015) and Marlier et al. (2012). In addition, both Varghese et al. (2011) and the Schaap et al. (2015) indicate that using approximately 0.5deg*

*resolution models (the resolution used in this study) for air quality is appropriate and useful.*

*We believe that this is a study of both atmospheric composition and air quality, and since we make important conclusions with regard to air quality, we feel the title accurately represents the work.*

*We considered only the changes in $O_3$ for the air quality standards because the relative changes in $O_3$ were much larger than the changes in any other WHO standard species, including particulate matter. This is due to the substantial background PM concentration associated with fires in the region. Many of these fires are used to clear land for oil palm plantations. This impact has been studied further in Marlier et al. (2015).*

*We clarify this in our manuscript on P11 Lines 24-27*

*The metric of number of days in exceedance is one that is commonly used in the atmospheric chemistry community for models of similar (and coarser) resolution, e.g.: Fiore et al. (2002), Parrish et al. (2010), Leibensperger et al. (2008), Lin et al. (2001), Van Loon et al. (2007), and Marlier et al. (2012).*

The temporal resolution of the quoted changes in atmospheric composition is also not sufficient for air quality assessments. While annual limits are given for some pollutants (although mostly in terms of accumulated exposure), daily 8-hour and peak 1-hour exposure is the more normal metric considered. Presenting changes in annual average concentrations is therefore inappropriate in the context of air quality.

*We presented our results as general long-term averages, followed by a metric-relevant analysis of daily maximum 8-hour average surface $O_3$ for urban air quality. This is similar to the way data is presented in Kim et al. (2015) and Marlier et al. (2012).*

It appears that GEOS-Chem was driven with meteorology for a single year (2006). The authors report that there was no substantial difference in projected changes in atmospheric composition between seasons. This is in contrast to the findings reported by Ashworth et al., 2012, and seems odd given that SE Asia is a monsoon-influenced region. That, plus the high level of fires reported for 2006, suggests that it may not have been a "typical" or representative year. Did the authors give any consideration to the inter-annual variability of their findings?

*We took this comment into consideration, and completed simulations using 2007 and 2008 meteorology and emissions. These results indicate that the absolute magnitude of changes presented in this work are not highly sensitive to the choice of model year. The relative changes are more sensitive to the choice of model year, wherein the high amount of fires in 2006 lead to more modest relative changes. We have added a sentence on P4 L8-9 in the manuscript:*

*"Additional simulations using emissions and meteorology from 2007 and 2008 indicate that the choice of model year does not substantially influence the results of this work."*

A future scenario set in 2020 seems rather limited in scope given that it is now 2016. It would have been interesting to assess how the LUC might combine with future changes in climate and air quality in the region with a longer-term scenario.

*In light of both Reviewers comments, we have changed the way we discuss the 2020 projections throughout the manuscript. To focus on the important notion that it is a near-term pessimistic future, and not a prediction of the exact distribution, we have added several sentences at the end of section 2.2:*

*"It is important to note that the 2020 distribution used here is the best estimation of a pessimistic future, and may not be an accurate prediction for the specific year 2020. It is meant to represent a realistic near-term scenario, and for this reason we refer to it from here on as the "future" distribution."*

*Throughout the paper, we now typically refer to "near-term future" rather than 2020. We agree that it would also be quite interesting to investigate how future air-quality and climate in the region interact with this land use change. However, we focus here only on near-term changes to the oil palm distribution, and the resulting influence on biosphere-atmosphere fluxes. Because of this, that specific climate analysis was considered out of the scope of this work.*

**Other**
The analysis is limited with changes in atmospheric composition given almost entirely in terms of changes in annual averages. On the whole, presentation of results is limited to a series of virtually identical figures. As most of the changes are spatially similar there seem an unnecessary number of figures. They do highlight the issue of model resolution quite clearly. Pugh et al., 2013 identified SE Asia as a region in which model spatial resolution is particularly important for atmospheric chemistry modeling which also appears not to have been considered by the authors.

*The figures are presented in a similar way to facilitate comparisons among them and with the satellite analysis in section 4. We also feel that the number of figures chosen allows for the best interpretation and reproducibility of these results in context with other studies.*

*We do in fact address issues related to resolution on P8L19 and P10 L21 in discussing the disagreement between our model and experimental data. However, we do not consider the model resolution to be an issue for the validity of our results, for the reasons and citations listed in the previous responses.*

*Pugh et al. (2013) demonstrate that using a 0.1˚x0.1˚ model resolution is far superior than using a 2˚x2˚ model over Southeast Asia. They further recommend that an*

*effective way to deal with high model uncertainty is to use "higher resolution land cover data, even when paired with coarser meteorological data". The model resolution we use is 0.5°x0.667°, significantly better than 2°x2°. Furthermore, we use a higher resolution land cover data (0.23°x0.31° resolution) as recommended by Pugh et al. (2013).*

The choice of color scale for Figure 9 is poor. It is virtually impossible to make out the outline of the islands when this is printed out. Using white for a ratio of unity would seem a more sensible way to show the limited extent of the impact.

*Thank you for this suggestion. The color scale has been changed.*

Isoprene emissions are not usually given in units of atoms C cm-2 s-1 in the context of a regional modeling.

*We have changed the units to $\mu mol\ C\ m^{-2}\ hr^{-1}$ for consistency with other work (Guenther et al. 2012).*

**References**
Ashworth, K., G. Folberth, C. N. Hewitt, and O. Wild. "Impacts of near-Future Cultivation of Biofuel Feedstocks on Atmospheric Composition and Local Air Quality." Atmos. Chem. Phys. 12, no. 2 (January 19, 2012): 919–39. doi:10.5194/acp-12-919-2012.

Fiore, A.M., D.J. Jacob, I. Bey, R.M. Yantosca, B.D. Field, A.C. Fusco, and J.G. Wilkinson, Background ozone over the United States in summer: Origin,trend, and contribution to pollution episodes, J. Geophys. Res., 107 (D15)

Fowler, D., E. Nemitz, P. Misztal, C. Di Marco, U. Skiba, J. Ryder, C. Helfter, et al. "Effects of Land Use on Surface-Atmosphere Exchanges of Trace Gases and Energy in Borneo: Comparing Fluxes over Oil Palm Plantations and a Rainforest." Philosophical Transactions of the Royal Society B: Biological Sciences 366, no. 1582 (November 27, 2011): 3196–3209. doi:10.1098/rstb.2011.0055.

Guenther, A. B., X. Jiang, C. L. Heald, T. Sakulyanontvittaya, T. Duhl, L. K. Emmons, and X. Wang. "The Model of Emissions of Gases and Aerosols from Nature Version 2.1 (MEGAN2.1): An Extended and Updated Framework for Modeling Biogenic Emissions." Geosci. Model Dev. 5, no. 6 (November 26, 2012): 1471–92. doi:10.5194/gmd-5-1471-2012.

Kim, P. S., D. J. Jacob, L. J. Mickley, S. Koplitz, M. E. Marlier, R. DeFries, S. S. Myers, B. N. Chew, and Y. H. Mao Sensitivity of population smoke exposure to fire locations in Equatorial Asia, Atmos. Environ., 102, 11-17, 2015.

Leibensperger, E. M., L. J. Mickley, D. J. Jacob, Sensitivity of U.S. air quality to mid-latitude cyclone frequency and implications of 1980-2006 climate change, Atmos. Chem. Phys., 8, 7075-7086, 2008

Lin, C.-Y. C., D.J. Jacob, and A.M. Fiore, Trends in exceedances of the ozone air quality standard in the continental United States, 1980-1998, Atmos. Environ., 35, , 3217-3228, 2001]

Marlier, Miriam E., Ruth S. DeFries, Apostolos Voulgarakis, Patrick L. Kinney, James T. Randerson, Drew T. Shindell, Yang Chen, and Greg Faluvegi. "El Nino and Health Risks from Landscape Fire Emissions in Southeast Asia." Nature Climate Change 3, no. 2 (February 2013): 131–36. doi:10.1038/nclimate1658.

Marlier, Miriam E., Ruth S. DeFries, Patrick S. Kim, David L. A. Gaveau, Shannon N. Koplitz, Daniel J. Jacob, Loretta J. Mickley, Belinda A. Margono, and Samuel S. Myers. "Regional Air Quality Impacts of Future Fire Emissions in Sumatra and Kalimantan." Environmental Research Letters 10, no. 5 (2015): 054010. doi:10.1088/1748-9326/10/5/054010.

Parrish, D. D., K. C. Aikin, S. J. Oltmans, B. J. Johnson, M. Ives, and C. Sweeny. "Impact of Transported Background Ozone Inflow on Summertime Air Quality in a California Ozone Exceedance Area." Atmos. Chem. Phys. 10, no. 20 (October 27, 2010): 10093–109. doi:10.5194/acp-10-10093-2010.

Pugh, T. A. M., K. Ashworth, O. Wild, and C. N. Hewitt. "Effects of the Spatial Resolution of Climate Data on Estimates of Biogenic Isoprene Emissions." Atmospheric Environment 70 (May 2013): 1–6. doi:10.1016/j.atmosenv.2013.01.001.

Schaap, M., C. Cuvelier, C. Hendriks, B. Bessagnet, J. M. Baldasano, A. Colette, P. Thunis, et al. "Performance of European Chemistry Transport Models as Function of Horizontal Resolution." Atmospheric Environment 112 (July 2015): 90–105. doi:10.1016/j.atmosenv.2015.04.003.

Van Loon, M., Robert Vautard, M. Schaap, Robert Bergström, Bertrand Bessagnet, J. Brandt, P. J. H. Builtjes et al. "Evaluation of long-term ozone simulations from seven regional air quality models and their ensemble." Atmospheric Environment 41, no. 10 (2007): 2083-2097.

Varghese, Saji, Baerbel Langmann, Darius Ceburnis, and Colin D. O'Dowd. "Effect of horizontal resolution on meteorology and air-quality prediction with a regional scale model." Atmospheric Research 101, no. 3 (2011): 574-594.

Warwick, N. J., A. T. Archibald, K. Ashworth, J. Dorsey, P. M. Edwards, D. E. Heard, B. Langford, et al. "A Global Model Study of the Impact of Land-Use Change in Borneo

on Atmospheric Composition." Atmospheric Chemistry and Physics 13, no. 18 (September 16, 2013): 9183–94. doi:10.5194/acp-13-9183-2013.

Wesely, M. L, and B. B Hicks. "A Review of the Current Status of Knowledge on Dry Deposition." Atmospheric Environment 34, no. 12–14 (2000): 2261–82. doi:10.1016/S1352-2310(99)00467-7.

---

## Author Response (AR1)

We thank the referees for their consideration of our manuscript. Below are our responses to each of the comments, including the proposed changes to our revised manuscript.

**Reviewer 1 Responses** General Comments**

1. I wonder if anything can be said about the credibility of the year 2020 projections since we are only four years away from 2020. There are some data available on forest area for 2015 from the FAO (http://www.fao.org/3/a-i4808e.pdf) which show about the same rate of forest loss in Indonesia between 2010-2015 as in 2005-2010, although not a perfect proxy for palm oil expansion in the two specific regions considered in this study.

The reviewer raises a good point. The projections from Marlier et al. (2015a) and Austin et al. (2015) that we use are specifically for 2020, however, given uncertainties in these trajectories, we have changed the way we discuss the 2020 projections throughout the manuscript. To focus on the important notion that it is a near-term future, and not a precise prediction, we have added several sentences at the end of section 2.2:

"It is important to note that the 2020 distribution used here is the best estimation of a near-term future wherein large increases in oil palm plantations continue to occur. The distribution may not be an accurate prediction for the specific year 2020. It is meant to represent a realistic near-term scenario, and for this reason we refer to it from here on as the "future" distribution."

Throughout the paper, we now typically refer to "near-term future" rather than 2020.

2. It would be helpful to include the locations and names of the major urban areas referred to in the text in one of the early figures, possibly figure 1a, especially given that their relative locations are important for palm oil plantation impacts

Thank you for this suggestion. A figure and descriptive sentence was added in section 2.1, the model description:

"A map of the Asian domain, and the region of particular interest to this study is shown in Figure 1."

**Minor Comments**

Pg 4, Line 3: Can you comment on how the high fire emissions and dry conditions associated with the beginning of an el nino event in late 2006 may or may not impact your annual average results?

We took this comment into consideration, and completed simulations using 2007 and 2008 meteorology and emissions. These results indicate that the absolute magnitude of changes presented in this work are not highly sensitive to the choice of model year. The relative changes are more sensitive to the choice of model year, wherein the high

amount of fires in 2006 lead to more modest relative changes. We have added a sentence on P4 L8-9 in the manuscript:

**"Additional simulations using emissions and meteorology from 2007 and 2008 indicate that the choice of model year does not substantially influence the results of this work."**

Pg 5, Line 4: I think the "native forest" here refers not to a new PFT but to broadleaf evergreen trees as mentioned later in the sentence, this could be made more clear so the reader understands exactly what is being changed. The need to reduce the isoprene emission factor by 4 times over the default seems like it could be a major issue for global studies given that this forest PFT is dominant in the region, topic for another study maybe.

The reviewer is correct, we have added an explanatory phrase to the text: "(considered to be broadleaf evergreen tropical trees)"

The reduction in isoprene emission factors over broadleaf evergreen trees is confined to only over Southeast Asia. As such, it is likely not a major issue for global studies.

Pg 5, Line 9: Delete "that"

Done.

Pg 6, Lines 12-14: Does the palm oil PFT always replace the tropical forest PFT or does it replace all PFTs in a grid cell, including crops and grasses, based on the original fraction of the PFTs? It is suggested later on in the text that other land cover types besides forest are being converted to palm oil (Pg 8, Line 2) but it would be helpful to have this made clear on Pg 6.

*Clarified in the text to: "fractional coverage of all pre-existing vegetation classes from the base land map are reduced accordingly."*

Pg 8, Line 26-27: Is the 10% mentioned here also relative to the Modern Palm simulation?

Added "relative to the Modern Palm simulation" to the end of that sentence.

Pg 12, Lines 10-13: Does the overlap in the "Palm" and "Ocean" filtering regions dilute the palm oil signal in the results shown in figures 14 and 15?

No it does not. Only satellite measurements over the given regions were selected from within the larger bounding boxes. We have modified the figure to clarify this.

Pg 13, Lines 18-19: This sentence was a bit unclear to me and I think the rest of the paragraph stands well enough on its own such that this sentence could probably be deleted.

**The sentence has been deleted.**

Pg 14, Line 1: The 30% increase in ozone – is this relative to the no-palm world, or to 2010, and is this the palm oil plantation contribution alone or does this include increases in other sources of ozone?

*This is relative to the no palm simulation, and is only the plantation contribution. The sentence has been altered to be more clear:*

*"If the oil palm crop expansion continues unabated, near-term future ozone concentrations in urban regions could be up to 30% higher (compared to the no palm scenario) due to the plantations alone."*

**Reviewer 2 Responses**

**General Comments**

The authors present a study of land conversion to oil palm plantations in SE Asia, although this is limited to Indonesian Borneo and Sumatra. They use the GEOS-Chem atmospheric chemistry transport model to investigate how changes in land cover affect emissions of volatile organic compounds (specifically isoprene) and the impacts this has on atmospheric composition in the region. They find that increasing the area of oil palm plantations increases isoprene emissions and hence ozone and aerosol concentrations over most of the region.

While this is clearly a topic of interest, and one that is central to the scope of ACP, the research presented here is not sufficiently novel in my opinion to warrant publication at this time.

We appreciate the reviewer's in depth analysis and comments on our work, however we disagree with several key points made by the reviewer. We note these throughout the responses.

We would also like to clarify the reviewer's characterization that that the work is "limited to Indonesian Borneo and Sumatra". We include the oil palm distribution across all of Malaysia as well, thus capturing >90% of all global oil palm agriculture. For the future scenario sensitivity, we only change the distribution of oil palm over Sumatra and Kalimantan due to a lack of information on the future distribution of palm throughout Malaysia. We believe that this is an acceptable sensitivity test because, as stated in our manuscript, the majority of land across both Indonesia and Malaysia available for oil palm plantation expansion is on Sumatra and Kalimantan.

**Novelty**

I do not feel that the work presented here is sufficiently novel in scope or methodology to mark an advance on previous work. Other than the use of different land use change maps the simulations do not differ from previous studies nor does the analysis of the model output extend beyond what has been considered before. We have updated the text to further emphasize the novelty of the work (described in the detailed points below).

We further note that the reviewer did not discuss the novel satellite analysis presented in Section 4, wherein we describe how and why current observing systems are not capable of detecting the air quality impacts of the massive land use change signature of oil palm. Our conclusions are critically important for implementation of observing systems of the future. If we cannot observe the impacts of the massive change in biogenic emissions associated with oil palm plantations based on current satellite measurement capabilities, we are likely to struggle to detect the impacts of any forest to agriculture land use conversions in the future.

The authors introduce a new oil palm-specific plant functional type (PFT) into GEOS- Chem, adapting isoprene emission factor and LAI of the existing tropical broadleaf evergreen tree PFT using data from the OP3 field campaign. This is the exact approach taken by Ashworth et al., 2012 and Warwick et al., 2013, which built on earlier investigations of LUC (although not specifically due to oil palm) by e.g. Wiedinmyer et al., 2006; Lathiere et al., 2006; Ganzeveld et al., 2010.

*Our implementation of the oil palm-specific plant functional type differs from both Ashworth et al. (2012) and Warwick et al. (2013) in several key ways:*

Ashworth et al. (2012) scaled the isoprene emissions of only broadleaf evergreen trees within a given grid box based on the fraction of oil palm expected within that grid box. The scaling factor used was a ratio of measured emission factors for oil palm and broadleaf evergreen trees: 50/35. For this work, we reduce the fraction of all vegetation in a grid box proportionally, not just broadleaf evergreen trees. This accounts for the fact that oil palm is not only farmed in regions where the natural rainforest has been removed. We additionally directly calculate the isoprene emissions through the MEGANv2.1 algorithm using the measured basal isoprene emission factors from OP3. By using the MEGAN algorithm in conjunction with the measured emission factors, we are able to more robustly estimate oil palm emissions outside of the timeframe of the OP3 field campaign (accounting for seasonal temperature differences, PAR, LAI, etc.) This was not explicitly accounted for in Ashworth et al. (2012) or Warwick et al. (2013).

Warwick et al. (2013) explored a future scenario where the entire island of Borneo was covered with oil palm vegetation, and replaced the MEGAN emissions algorithm with emissions measured during the OP3 field campaign. They acknowledge that this is "obviously an extreme situation", but that is useful for exploring a certain air quality trajectory. We use a more realistic land map for both modern and near-term future oil palm distributions, including the distribution of oil palm on Sumatra and the Malay Peninsula.

We clarify some of these differences in our manuscript on P2 Lines 31-34. And P5 L9-11 The authors demonstrate that changes in oil palm distribution alter isoprene emissions (and hence concentrations) and thence concentrations of  $O_3$  and SOA. This is not a new finding (Ashworth et al., 2012; Warwick et al., 2013, and many other studies showing that different land cover affects atmospheric composition via changes in bio- genic VOC emissions, e.g. Guenther et al., 2006; Arneth et al., 2011; and those listed above). The authors' results differ only in terms of distribution and scale.

We agree that the finding that changes in isoprene emissions can change  $O_3$  and SOA concentrations is not unique to this study. We argue that the differences that we model in terms of distribution, scale, and magnitude are important enough to consider this work novel. In particular, our simulations explore the integrated effects of changing emissions AND deposition and are at higher resolution (0.5x0.67) than Ashworth et al. (2012) and Warwick et al. (2013). Previous studies also do not explore the impact of oil palm on densely populated regions, such as the Malay Peninsula, where we see the largest surface O3 increases.

Including changes in NOx emissions concomitant to the changes in land cover is not new, and in fact the study here does not go as far as Ashworth et al., 2012 who included sensitivity tests with and without NOx emissions associated with processing, nor Warwick et al., 2013 who included changes in soil NOx emissions associated with periodic fertilization of oil palm plantations.

We agree with the reviewer: the treatment of NOx emissions is not a novel aspect of our study. Given the uncertainties associated with fertilization (as discussed for the OP3 field campaign by Fowler et al. (2011)) and processing emissions, changes in NOx are not a focus of this study. More information on additional NOx sources from fertilization or processing is needed to realistically explore this. However, we note that our satellite analysis in Section 4 confirms that we are not missing major palm-related sources of NOx in our simulation, suggesting that these sources may indeed be modest.

Changes in O3 deposition have also been included in many previous studies of LUC (e.g. Ganzeveld et al., 2010; Ashworth et al., 2012).

We agree that changes in  $O_3$  deposition have been included in studies of land use change before. However, the simultaneous exploration of both emissions and deposition changes related to oil palm had not been studied before. Warwick et al (2013) and Ashworth et al. (2012) both explored separate sensitivity studies to show the potential influence of deposition. Warwick et al (2013) doubled all deposition velocities, to test their model sensitivity. Ashworth et al. (2012) perform several short sensitivity studies of deposition, and conclude that the changes in deposition are likely to increase  $O_3$  concentrations. This is at odds with our work, where we show the opposite through a complete simulation of the depositional changes. The Ashworth et al. (2012) result is likely due to the fact that they used a higher biomass density for the natural vegetation than for oil palm plantations, and only allowed oil palm to replace the tropical broadleaf evergreen trees. This reduction in biomass density led to a reduction in deposition. Our work uses a dry deposition scheme (Wesely and Hicks, 2000) that does not use biomass density as a parameter, but instead uses LAI. This is likely a more realistic parameter due to the heavy dependence on stomatal and cuticular deposition in heavily forested regions. The observationally-derived land cover inputs used in this work show that oil palm plantations have higher LAI than most of the natural rainforest and other land types across Southeast Asia. This leads to the increased deposition in our work.

**We added text to P6 L4-8 and P6 L27-33**

Furthermore, since the publication of the OP3 data that the authors cite here, further data have been presented reporting high emissions of other VOC from oil palm (e.g. methyl chavicol (estragole) and toluene, Mizstal et al., 2010; 2011; 2015).

The reviewer correctly points out that the oil palm plantations have higher emissions of estragole and toluene, however their chemical transformations (and thus impact on  $O_3$  and SOA) are not well constrained, and Guenther et al. (2012) indicate that emissions of these species are relatively unimportant, particularly as compared to isoprene. As a result, we do not include these species in our analysis and do not believe that this omission significantly impacts our results. We add a sentence to P5 L18-20 discussing this.

Higher than expected deposition of a number of other compounds has also been reported (e.g. Karl et al., 2009; Nguyen et al., 2014) and yet this does not appear to have been considered by the authors who refer to the reactivity of ozone as an additional reason to focus on its deposition. I would also be interested to know if GEOS-Chem partitions dry deposition between stomatal and non-stomatal routes in line with e.g. Fares et al., 2012; 2013; 2014; Simpson et al., 2012.

The changes in land cover do change the deposition of other compounds in GEOS-Chem. We focus on  $O_3$  due to the significance of  $O_3$  on regional air quality, and because its reactivity makes it a good candidate for describing the changes in deposition related to oil palm expansion. The magnitude of changes in deposition for all other species is nearly always less than 10%, and never more than 15%. The dry deposition module in GEOS-Chem does consider deposition through both stomatal and nonstomatal pathways, as outlined in Wesely and Hicks (2000), but only calculates one net surface sink.

**Methodology**

The spatial resolution of the model (0.5x0.66667deg) is too coarse for studying air quality (see e.q. Gego et al., 2005; Varghese et al., 2011; Schaap et al., 2015). In my view, this is a study of impacts on atmospheric composition rather than air quality and should be so described (i.e. in the title and text). Further, the authors only

demonstrate how the projected changes in O3 concentrations relate to recognized WHO air quality standards although they discuss changes in formaldehyde (not a regulatory air pollutant), NOx and SOA as well. And yet, premature mortality and morbidity associated with particulate matter is almost an order of magnitude higher than for O3. I also find the choice of metric odd; the number of exceedance days is a threshold metric (i.e. a consideration of "extreme" conditions) which is likely to be poorly represented by a coarse resolution model.

This model resolution has commonly been used to study air quality in Southeast Asia, including: Kim et al. (2015) and Marlier et al. (2012). In addition, both Varghese et al. (2011) and the Schaap et al. (2015) indicate that using approximately 0.5deg resolution models (the resolution used in this study) for air quality is appropriate and useful.

We believe that this is a study of both atmospheric composition and air quality, and since we make important conclusions with regard to air quality, we feel the title accurately represents the work.

We considered only the changes in  $O_3$  for the air quality standards because the relative changes in  $O_3$  were much larger than the changes in any other WHO standard species, including particulate matter. This is due to the substantial background PM concentration associated with fires in the region. Many of these fires are used to clear land for oil palm plantations. This impact has been studied further in Marlier et al. (2015b).

We clarify this in our manuscript on P11 Lines 24-27

The metric of number of days in exceedance is one that is commonly used in the atmospheric chemistry community for models of similar (and coarser) resolution, e.g.: Fiore et al. (2002), Parrish et al. (2010), Leibensperger et al. (2008), Lin et al. (2001), Van Loon et al. (2007), and Marlier et al. (2012).

The temporal resolution of the quoted changes in atmospheric composition is also not sufficient for air quality assessments. While annual limits are given for some pollutants (although mostly in terms of accumulated exposure), daily 8-hour and peak 1-hour exposure is the more normal metric considered. Presenting changes in annual average concentrations is therefore inappropriate in the context of air quality.

We presented our results as general long-term averages, followed by a metric-relevant analysis of daily maximum 8-hour average surface  $O_3$  for urban air quality. This is similar to the way data is presented in Kim et al. (2015) and Marlier et al. (2012).

It appears that GEOS-Chem was driven with meteorology for a single year (2006). The authors report that there was no substantial difference in projected changes in atmospheric composition between seasons. This is in contrast to the findings reported by Ashworth et al., 2012, and seems odd given that SE Asia is a monsooninfluenced region. That, plus the high level of fires reported for 2006, suggests that it may not have been a "typical" or representative year. Did the authors give any consideration to the inter-annual variability of their findings?

We took this comment into consideration, and completed simulations using 2007 and 2008 meteorology and emissions. These results indicate that the absolute magnitude of changes presented in this work are not highly sensitive to the choice of model year. The relative changes are more sensitive to the choice of model year, wherein the high amount of fires in 2006 lead to more modest relative changes. We have added a sentence on P4 L8-9 in the manuscript:

"Additional simulations using emissions and meteorology from 2007 and 2008 indicate that the choice of model year does not substantially influence the results of this work."

A future scenario set in 2020 seems rather limited in scope given that it is now 2016. It would have been interesting to assess how the LUC might combine with future changes in climate and air quality in the region with a longer-term scenario.

In light of both Reviewers comments, we have changed the way we discuss the 2020 projections throughout the manuscript. To focus on the important notion that it is a near-term pessimistic future, and not a prediction of the exact distribution, we have added several sentences at the end of section 2.2:

"It is important to note that the 2020 distribution used here is the best estimation of a pessimistic future, and may not be an accurate prediction for the specific year 2020. It is meant to represent a realistic near-term scenario, and for this reason we refer to it from here on as the "future" distribution."

Throughout the paper, we now typically refer to "near-term future" rather than 2020. We agree that it would also be quite interesting to investigate how future air-quality and climate in the region interact with this land use change. However, we focus here only on near-term changes to the oil palm distribution, and the resulting influence on biosphere-atmosphere fluxes. Because of this, that specific climate analysis was considered out of the scope of this work.

**Other**

The analysis is limited with changes in atmospheric composition given almost entirely in terms of changes in annual averages. On the whole, presentation of results is limited to a series of virtually identical figures. As most of the changes are spatially similar there seem an unnecessary number of figures. They do highlight the issue of model resolution quite clearly. Pugh et al., 2013 identified SE Asia as a region in which model spatial resolution is particularly important for atmospheric chemistry modeling which also appears not to have been considered by the authors. The figures are presented in a similar way to facilitate comparisons among them and with the satellite analysis in section 4. We also feel that the number of figures chosen allows for the best interpretation and reproducibility of these results in context with other studies.

We do in fact address issues related to resolution on P8L19 and P10 L21 in discussing the disagreement between our model and experimental data. However, we do not consider the model resolution to be an issue for the validity of our results, for the reasons and citations listed in the previous responses.

Pugh et al. (2013) demonstrate that using a 0.1°x0.1° model resolution is far superior than using a 2°x2° model over Southeast Asia. They further recommend that an effective way to deal with high model uncertainty is to use "higher resolution land cover data, even when paired with coarser meteorological data". The model resolution we use is 0.5°x0.667°, significantly better than 2°x2°. Furthermore, we use a higher resolution land cover data (0.23°x0.31° resolution) as recommended by Pugh et al. (2013).

The choice of color scale for Figure 9 is poor. It is virtually impossible to make out the outline of the islands when this is printed out. Using white for a ratio of unity would seem a more sensible way to show the limited extent of the impact.

**Thank you for this suggestion. The color scale has been changed.**

Isoprene emissions are not usually given in units of atoms C cm-2 s-1 in the context of a regional modeling.

We have changed the units to  $\mu$ mol C m-2 hr-1 for consistency with other work (Guenther et al. 2012).

[revised manuscript text omitted]

Figure & shows that the simulated response of surface NOx to the oil palm expansion is very small. In principle, this response is influenced by changes to deposition, soil NOx emissions, and isoprene fluxes. Given the modest difference in deposition and soil NOx emissions, the dominant impact is the elevated concentrations of isoprene. Additional isoprene leads to more

9

Sam Silva 6/10/2016 8:52 AM Deleted: %.

Sam Silva 6/10/2016 8:52 AM Deleted: 6

Sam Silva 6/10/2016 8:52 Al Deleted: 7 conversion of NO to NO2, and therefore increases the formation of HNO3, leading to a net loss of NOx. This effect is only apparent across the southern Malay Peninsula, a region with high surface NOx concentrations, due in large part to significant anthropogenic activity. These changes are typically less than 0.1 ppbv, on the order of 5% decreases. The Future oil palm simulation shows similar decreases in the surface NOx response. These decreases are as large as 1 ppbv over Sumatra, a 5%

5 drop. There is a decrease in NOx across Kalimantan on the order of  $\sim$ 0.5 ppbv related to the same chemistry. In reality, these changes may be dwarfed by the impact of anthropogenic emissions of NOx associated with production and processing facilities as well as oil palm fertilization; these changes are highly uncertain, and have not been included here.

The introduction of oil palm and the resulting increase in concentrations of isoprene can lead to changes in concentrations of
 ozone through VOC-NOx chemistry. At the same time, the increase in the deposition velocity of ozone leads to a shorter average lifetime, which decreases concentrations. In our modeled responses we see both of these signatures across SEA.
 Figure 2 shows that the surface ozone response to Modern Palm is most prominent over the southern Malay Peninsula (up to 4 ppbv), with changes over northeastern Borneo and Sumatra as well. Over the Malay Peninsula and Sumatra, a region not sampled during OP3, surface ozone concentrations increase by up to 26% (3-4 ppbv) due to palm expansion. Ozone

- 15 formation is enhanced in these regions, where additional isoprene emissions combine with  $NO_x$  rich air near the major urban centers. Surface ozone increases in northeastern Borneo are on the order of 2 ppbv, located in the near vicinity of the oil palm plantations. These results differ spatially from Warwick et al. (2013), likely due to the substantially different land maps used for oil palm emissions of VOCs over Borneo. Hewitt et al. (2009) did not observe a change in surface  $O_3$ concentrations due to oil palm at all over northeastern Borneo. Much of the discrepancy between our results and the Hewitt
- 20 et al. (2009) observations can likely be explained by sampling and the different spatial resolution of the measurements and the model. The  $0.5^{\circ}x0.666^{\circ}$  grid box resolution used in this study is on the order of the entire study region for OP3.

Adding oil palm plantations usually increases the LAI (Figure  $\frac{1}{2}$ ), leading to an increased depositional velocity (Figure  $\frac{1}{2}$ ), which ultimately results in an increased sink of O3. However, this is generally counteracted by the large increase in isoprene

[revised manuscript text omitted]

Sam Silva 6/10/2016 8:52 AM Deleted: 6

Sam Silva 6/10/2016 8:52 AM Deleted: 13 Sam Silva 6/10/2016 8:52 AM Deleted: 13 Sam Silva 6/10/2016 8:52 AM Deleted: 14

Sam Silva 6/10/2016 8:52 AM Deleted: 12

Sam Silva 6/10/2016 8:52 AM Deleted: 15 Sam Silva 6/10/2016 8:52 AM Deleted: 12 forest difference of  $10^{14}$  molecules cm-2 (~25%) agrees with our modeled analysis, which shows surface concentrations that differ by ~29% in this region, related to elevated anthropogenic NOx emissions. This suggests that we are not missing major palm-related sources of NOx emissions (fertilizer or industrial processing) in our simulation. The constant increase in retrieved NO2 concentrations over all three regions is consistent with Geddes et al. (2015b), who show a broad increasing

5 trend in NO2 across all of northern Borneo, possibly due to warmer surface temperatures, and transport from urban regions.

The satellite-derived signal in tropospheric ozone from oil palm development near Kuala Lumpur is not apparent against the background of urban development. Even though there are significant changes in the local ozone concentrations, too many confounding sources exist to identify the oil palm signal. Fires in SEA dominate the measured AOD, with an additional

10 contribution from urban sources (Cohen and Lecoeur, 2015). Since the AOD measurement is a net observation of extinction from all aerosols at all altitudes, detecting changes in surface-level SOA is not straightforward. It is therefore challenging to identify the impact of oil palm expansion on air quality in SEA with the current constellation of polar-orbiting satellites.

In light of this result, it is important to consider the monitoring capabilities of future observing systems, such as the North
 American geostationary mission TEMPO (Chance et al. 2013). Geostationary observations offer more frequent sampling of the diurnal cycle, which may enable a separation of source signatures (e.g. urban, fire, biogenic, etc.) and a better identification of perturbations associated with land use change. In terms of HCHO measurements, TEMPO has a similar precision to that of the OMI sensor (1016 molecules cm-2). However, TEMPO will have much smaller spatial footprint (8x4.5)

- km) as compared to OMI (14x24 km), and will sample HCHO three times daily, as opposed to the once daily measurements
   from OMI. The combination of these factors will likely make changes in the emissions of HCHO and its precursors more
- detectable, if they occur within the geostationary viewing field. Our results indicate that the instrument precision and footprint size are the most important limiting factor in detection of the perturbation associated with oil palm plantations with the current suite of satellite observations.

**5. Conclusions**

- 25 In this study, we simulate the impact of recent and near-term projected oil palm expansion across SEA on air quality. We go beyond previous work by consistently treating the impact of land use change on a suite of land-atmosphere exchange processes relevant to atmospheric chemistry. Our simulations suggest that oil palm plantation expansion in the region has had a significant impact on air quality. As oil palm expansion continues, the potential impact on surface O3 concentrations is significant. The predicted ozone changes are largely due to increasing isoprene emissions. Locally however, increases in
- 30 depositional velocities counteract these elevated emissions. If the oil palm crop expansion continues unabated, near-term future ozone concentrations in urban regions could be up to 30% higher (compared to the no palm scenario) due to the plantations alone. Exposure to ozone is a significant cause of premature mortality, responsible for more than 200,000 deaths

14

**Sam Silva 6/10/2016 8:52 AM**

Sam Silva 6/10/2016 8:52 AM Deleted: (2020)

Sam Silva 6/10/2016 8:52 AM **Deleted:** increase by Sam Silva 6/10/2016 8:52 AM **Deleted:** in 2020 
[revised manuscript text omitted]

---

## Author Response (AR2)

**Reviewer 2 Comments:**
I thank the authors for their detailed response to my concerns; they do not significantly alter my view of the manuscript as a whole. As the most novel aspect of the study is the analysis of the efficacy of current satellite observations in deducing air quality changes in the region I would suggest that this become the main focus of the report.

Replies to specific points made by the authors are shown below.

We thank the reviewer for their additional comments. We respectfully disagree with their overall evaluation, but have endeavored to address their specific concerns below. Our responses are shown in blue beneath their comments in black. Our previous responses to reviews are noted in italics where applicable.

Response to authors' comments:
* * *
To reviewer 1:
*"We … completed simulations using 2007 and 2008 meteorology and emissions. These results indicate that the absolute magnitude of changes presented in this work are not highly sensitive to the choice of model year. The relative changes are more sensitive to the choice of model year … We have added a sentence on P4 L8-9 in the manuscript: "Additional simulations using emissions and meteorology from 2007 and 2008 indicate that the choice of model year does not substantially influence the results of this work." "*

Please quantify the statement added to the manuscript so that the reader can judge how substantial the interannual variability.

The sentence was modified to:
"Additional simulations using emissions and meteorology from 2007 and 2008 indicate that the choice of model year does not substantially influence the results of this work, spatial patterns are all consistent and concentration changes are all less than 10%."
* * *
*"The reduction in isoprene emission factors over broadleaf evergreen trees is confined to only over Southeast Asia. As such, it is likely not a major issue for global studies."*

Can the authors provide a reference for this assertion that it is not of global importance? The OP3 campaign found that the emission factors used for broadleaf evergreen trees in MEGANv2.0, which were based on measurements in the Amazon, were a factor of 4 too high for SE Asia, a previously under-studied region. However the tropics as a whole, outside of Amazonia, could be considered under-studied. How confident are we of emission factors for African ecosystems for example? In which case, while the over- / under-estimation of emissions for other regions may differ from SE Asia uncertainty in emission factors could be a global issue.

We cannot comment on isoprene emissions from broadleaf evergreen trees outside of SEA. Our study is limited to this region, and the increase in basal EF that we employ is based solely on the OP3 observations over Borneo. To our knowledge there are no observational constraints on these emissions in other tropical regions (e.g. Africa). Our assertion that the difference is not of global importance relates only to how these emissions were implemented in our study (i.e. only over SEA). According to our model simulations, the reduction of isoprene emissions over SEA leads to a net isoprene emissions reduction of approximately 37 Tg/yr. This is well below 10% of the global isoprene emissions (Guenther et al., 2012) and is therefore not a substantial global perturbation. Considering that there are significant uncertainties in the estimation of global isoprene emissions; Guenther et al (2012) only report confidence to within a factor of two. It is likely that much of this uncertainty is related to poor emissions estimations due to under-studied regions. We therefore agree with the review that this remains an open (and important!) question, however it is outside of the scope of this study focused on SEA.

To reviewer 2:
*"We have updated the text to further emphasize the novelty of the work"*
The current work is now better contextualized in terms of previous studies.

Thank you; we appreciate the suggestion to expand this text.
* * *
*"We use a more realistic land map for both modern and near-term future oil palm distributions, including the distribution of oil palm on Sumatra and the Malay Peninsula."*

As stated in my previous review, this is the main difference between this and previous work, other than the satellite observations (see below).

We agree that this is a novel aspect of the work. Combined with the satellite analysis, as well as a modeling analysis that focuses on (a) the implications for urban air quality and (b) a deeper examination of the coincident changes in dry deposition and emissions, we strongly believe that our analysis is a novel and complementary addition to the existing literature.
* * *
*"Ashworth et al. (2012) scaled the isoprene emissions of only broadleaf evergreen trees within a given grid box based on the fraction of oil palm expected within that grid box. The scaling factor used was a ratio of measured emission factors for oil palm and broadleaf evergreen trees: 50/35. … We additionally directly calculate the isoprene*

*emissions through the MEGANv2.1 algorithm using the measured basal isoprene emission factors from OP3. By using the MEGAN algorithm in conjunction with the measured emission factors, we are able to more robustly estimate oil palm emissions outside of the timeframe of the OP3 field campaign (accounting for seasonal temperature differences, PAR, LAI, etc.) This was not explicitly accounted for in Ashworth et al. (2012) ...."*

It is my understanding that emissions in Ashworth et al. (2012) were also calculated on-line although using scaled emission factors (which are in themselves merely a scaling factor of emissions at non-standard conditions). Temperature and PAR effects were therefore explicitly accounted for, although changes in LAI would not have been included. I would expect seasonal changes in LAI for broadleaf evergreen trees and oil palm to be small however although that may not be the case in global models such as GEOS-Chem and HadGEM.
Please alter the amended text in Section 2.2 to reflect that the Ashworth et al. (2012) emissions were also calculated on-line as the added sentences imply off-line emissions were used without changes in meteorology being considered.

Thank you for this comment. Our initial response was worded poorly, and we agree that Ashworth et al. (2012) does indeed account for changes in PAR and Temperature. We agree that our text was not clear on this point and we have reworded the description on P5L10:
"Previous studies scaled modelled emissions of Broadleaf Evergreen Trees (Ashworth et al., 2012)…"
* * *
*"We further note that the reviewer did not discuss the novel satellite analysis presented in Section 4, wherein we describe how and why current observing systems are not capable of detecting the air quality impacts of the massive land use change signature of oil palm. Our conclusions are critically important for implementation of observing systems of the future. If we cannot observe the impacts of the massive change in biogenic emissions associated with oil palm plantations based on current satellite measurement capabilities, we are likely to struggle to detect the impacts of any forest to agriculture land use conversions in the future."*

If this is the novel aspect of the study presented in this manuscript it seems strange that it is given so little emphasis. Furthermore the "massive signature" of oil palm expansion in the region that the authors refer to has been deduced entirely through model simulations. The authors have not demonstrated that the model has been able to successfully simulate historical changes in air quality in the region and that model future projections are reliable. My understanding is that the OP3 campaign did not detect substantial differences in air quality above the oil palm plantations compared with native rainforest in spite of large differences in basal emission rates. This was ascribed to "NOx-limitation" and the authors deduced that in higher NOx regions air quality changes could be large. I agree that with our current understanding we would expect to see substantial changes in levels of secondary pollutants in the vicinity of urban areas downwind of the expansion but it has not been demonstrated. However, as suggested previously, I would urge the authors to re-frame the manuscript to focus on this aspect of their study as it is here that the novelty lies.

We agree with the reviewer that air quality observations over SEA would be invaluable in fully understanding the changes in air quality associated with oil palm plantations as well as to more robustly verify and validate model simulations. Given the paucity of such measurements (beyond the limited domain of the OP3 campaign in low NOx regions, as the reviewer suggests), we attempt to use satellite observations for this purpose in our work, and spend a substantial portion of our manuscript devoted to describing the issues regarding such an analysis. In light of the dearth of observations, we believe that the best tool for exploring these changes is a model such as the one used here.
Please see above for our comments on the originality of our study.

[revised manuscript text omitted]